# Finite Sample Analysis for Single-Loop Single-Timescale Natural Actor-Critic Algorithm

## Abstract

Natural actor-critic (NAC) methods have demonstrated remarkable effectiveness in various reinforcement learning problems. However, there remains a noticeable gap in the literature regarding the finite-time analysis of this practical algorithm. Previous theoretical investigations of actor-critic techniques primarily focused on the double-loop form, involving multiple critic steps per actor step, or the two-timescale form, which employs an actor step size much slower than that of the critic. While these approaches were designed for ease of analysis, they are seldom utilized in practical applications. In this paper, we study a more practical single-loop single-timescale natural actor-critic algorithm, where step sizes are proportional and critic updates with only a single sample per actor step. Our analysis establishes a finite sample complexity of $O(\epsilon^{-4})$, ensuring the attainment of the $\epsilon$-accurate global optimal point. To the best of our knowledge, we are the first to provide finite-time convergence with the global optimality guarantee for the single-loop single-timescale natural actor-critic algorithm with linear function approximation.

## 1 Introduction

Actor-Critic (AC) algorithms have achieved significant success and emerged as a popular approach in various reinforcement learning algorithms since their introduction (Konda & Borkar, 1999; Konda & Tsitsiklis, 1999). In the AC algorithm, the actor updates the policy by estimating the policy gradient (PG), a function of the Q-value corresponding to the policy. To accurately evaluate the Q-value, the AC algorithm employs the critic to track the value function. This design often reduces variance and accelerates convergence in practice, compared to using the Monte Carlo rollout to estimate the Q-value in the PG method. The Natural Actor-Critic (NAC) algorithm is a variant of AC, in which the actor adopts the Natural Policy Gradient (NPG) algorithm. NAC methods are widely used in practice, such as in trust region policy optimization (TRPO) and proximal policy optimization (PPO), and they often outperform AC in numerous applications.

Previous theoretical analyses for AC and NAC primarily focus on two variants based on the same concept. First, numerous studies consider double-loop variants, wherein the inner critic takes multiple update steps to accurately approximate the Q-value per actor step. The analysis for the actor and critic can be decoupled, reducing the problem to the policy estimation sub-problem in the inner loop and the policy gradient with the error sub-problem in the outer loop. It is important to note that this double-loop design is primarily for simplifying analysis and is seldom used in practice. Generally, it is sample inefficient compared to single-loop AC (NAC) as it requires accurate estimation of the Q-value function for the current policy.

The second variant involves two-timescale methods, wherein the actor's step size decays much slower than the critic's, with the ratio of their rates converging to zero as the iteration number approaches infinity. This design ensures that the actor always has an accurate Q-value estimation. Consequently, the analysis can be decoupled for the actor and the critic, akin to the double-loop variation analysis. It is important to note that two-timescale methods are not frequently employed in practice, as they necessitate artificially slowing down the actor, leading to inefficient sample complexity.

The theoretical analysis for the more practical single-loop single-timescale AC (NAC) presents a greater challenge. As the actor and the critic diminish at the same timescale, the decoupling analysis technique becomes inapplicable. It is necessary to control the actor's and critic's errors simultaneously, as they are deeply coupled due to the parallel update rule. Existing works analyze the

single-timescale Actor-Critic and demonstrate an $O\left(\epsilon^{-2}\right)$ sample complexity to reach the stationary point. These studies treat the two errors as components of an interconnected system. Analyzing the single-loop single-timescale NAC is more challenging than the Actor-Critic and remains unexplored. Since NAC employs the Fisher information matrix for the actor update, it is also essential to control the error of estimating the Fisher information matrix, which depends on the current policy. In the double-loop NAC, the Fisher information matrix can be approximated using multiple samples in the inner loop (Xu et al., 2020). However, in the single-loop single-timescale algorithm, only one sample is induced from the current policy and can be used to approximate the Fisher information matrix unbiasedly. A potential approach is to compute the Fisher information matrix as a linear combination of the estimation in the last iteration and the unbiased estimation of the current sample. Then, the estimating error can be controlled by the estimating error in the last round and the policy shift between iterations, which is related to the actor's and critic's errors. Consequently, it is necessary to simultaneously control these three estimating errors. This requirement makes the analysis for NAC more challenging.

## 1.1 CONTRIBUTION

- In this paper, we provide the first finite time sample complexity guarantee for the single-loop single-timescale NAC algorithm with linear function approximation, which is $O\left(\epsilon^{-2}\right)$ to find an $\epsilon$-approximate stationary point ($\|\nabla_\theta V(\theta)\|_2^2 < \epsilon +$ error) and $O\left(\epsilon^{-4}\right)$ to find an $\epsilon$-global optimal ($V^* - V(\theta) < \epsilon +$ error), where $V(\theta)$ is the value function of policy $\pi_\theta$, and $V^*$ is the value function of the optimal policy.

- To reduce the Fisher information matrix estimation's variance, we combine the previous estimation with the unbiased estimation using one sample. This update rule makes the estimation error related to the actor's and the critic's errors. Intuitively, when the actor takes a step small enough, the Fisher information matrix estimation is accurate since it does not change much between two iterations.

- The technical challenge in the analysis is that we need to simultaneously control the error of the actor update, the critic update, and the Fisher information matrix estimation. In the analysis of AC, Olshevsky & Gharesifard (2022) use the small gain analysis that bounds the critic error and the actor error with each other. The main difference and challenge in analyzing NAC is that we also need to bound the error for estimating the Fisher information matrix. Here we control these three errors together as an interconnected system and derive the final bound for the sample complexity.

## 2 RELATED WORK

The AC algorithm was initially proposed by Konda & Borkar (1999), which was later extended to the NAC algorithm by Kakade (2001). The asymptotic convergence of AC algorithms has been well-established in various settings (Kakade, 2001; Bhatnagar et al., 2009; Castro & Meir, 2010; Zhang et al., 2020). Recent research has focused on the finite-time convergence of AC methods. Yang et al. (2019) established global convergence of AC methods for solving the linear quadratic regulator (LQR) under the double-loop setting. Wang et al. (2019) investigated the global convergence of AC methods using neural network parameterization for both the actor and critic. Kumar et al. (2019) studied the finite-time local convergence of several AC variants with linear function approximation.

For the two-timescale AC algorithm, a study by Wu et al. (2020) demonstrated finite-time local convergence to a stationary point with a sample complexity of $O(\epsilon^{-2.5})$ for finite action spaces. Another work by Xu et al. (2020) investigated both local and global convergence for double-loop AC, achieving sample complexities of $O(\epsilon^{-2.5})$ and $O(\epsilon^{-4})$, respectively, under the discounted accumulated reward setting. Chen et al. (2022) established global convergence of two-timescale AC methods for solving LQR, utilizing a single sample to update the critic in each iteration.

For the single-timescale algorithm, Fu et al. (2020) considered the least-squares temporal difference update for the critic and obtained the optimal policy within the energy-based policy class for both linear function approximation and nonlinear function approximation using neural networks. Furthermore, Olshevsky & Gharesifard (2022); Chen et al. (2021); Chen & Zhao (2022) investigated the single-timescale AC in general MDP cases.

| | setting | paper | convergence rate |
|---|---|---|---|
| AC | double-loop | Wang et al. (2019) | $O(1/\epsilon^4)$ |
| | | Kumar et al. (2019) | $O(1/\epsilon^4)$ |
| | | Xu et al. (2020) | $O(1/\epsilon^2)$ |
| | single-loop two-timescale | Qiu et al. (2021) | $O(1/\epsilon^4)$ |
| | | Wu et al. (2020) | $O(1/\epsilon^{2.5})$ |
| | single-loop single-timescale | Chen et al. (2021) | $O(1/\epsilon^2)$ |
| | | Chen & Zhao (2022) | $O(1/\epsilon^2)$ |
| | | Olshevsky & Gharesifard (2022) | $O(1/\epsilon^2)$ |
| NAC | double-loop | Xu et al. (2020) | $O(1/\epsilon^3)$ |
| | single-loop two-timescale | Khodadadian et al. (2021) | $O(1/\epsilon^6)$ |
| | single-loop single-timescale | **this work** | $O(1/\epsilon^4)$ |

Table 1: Comparisons of settings and convergence rates with most related works.

## 3 PRELIMINARIES

This section introduces the basics of the discounted Markov decision process, natural policy gradient algorithm, and single-timescale NAC.

We consider a discounted Markov Decision Process (MDP) with finite states and actions, where $\mathcal{S}$ is the state space, $\mathcal{A}$ is the action space, and $P(s'|s, a)$ denotes the transition kernel that the current state $s$ transits to $s'$ after taking action $a$. Denote $r(s, a) \in [0, 1]$ as the reward given the state-action pair $(s, a)$. We will assume that we parameterize the set of policies by $\theta \in \Theta$ with $\pi_\theta(a|s)$ is the probability of choosing action $a$ in state $s$. Define the value-function $V_\theta(s) = \mathbb{E}_\theta \left[ \sum_{t=0}^\infty \gamma^t r_t(s_t, a_t) \mid s_0 = s \right]$, where the expectation $\mathbb{E}_\theta$ is taken over the Markov chain under the policy $\pi_\theta$. Define the Q-value as $Q_\theta(s, a) = \mathbb{E}_\theta \left[ \sum_{t=0}^\infty \gamma^t r_t(s_t, a_t) \mid s_0 = s, a_0 = a \right]$. Denote the distribution over the starting state as $\rho$, and we ca define the expected value function under policy $\pi_\theta$ as

$$V(\theta) = \mathbb{E}\left[V_\theta(s)\right] = \sum_s \rho(s) V_\theta(s).$$

Denote $p_{k,\theta}(s)$ the probability that the state transits to $s$ at step $k$ if policy $\pi_\theta$ is followed. Denote $\mu_\theta$ as the stationary state distribution induced by $\pi_\theta$. Following the policy gradient theorem, we can differentiate $V_\theta$ as:

$$\nabla_\theta V(\theta) = \sum_s \sum_{k=0,1,\dots} \gamma^k p_{k,\theta}(s) \sum_a \pi_\theta(a \mid s) Q_\theta(s, a) \nabla_\theta \log \pi_\theta(a \mid s)$$
$$= \sum_{s,a} \mu_\theta(s, a) Q_\theta(s, a) \nabla_\theta \log \pi_\theta(a \mid s).$$

This form allows us to apply a stochastic version of gradient descent. For the policy $\pi_{\theta_t}$ estimated at $t$, we can generate a random sample $(s_t, a_t)$ from the stationary distribution $\mu_{\theta_t}$ and update as

$$\theta_{t+1} = \theta_t - \beta_t Q_{\theta_t}(s_t, a_t) \nabla_\theta \log \pi_{\theta_t}(a_t \mid s_t).$$

For the natural policy gradient, it applies natural gradient descent, which is invariant to the parametrization of policies. Define $F(\theta_t) = \mathbb{E}_{\mu_{\theta_t}} \left[ \nabla_\theta \log \pi_{\theta_t}(a_t|s_t) \nabla_\theta \log \pi_{\theta_t}(a_t|s_t)^T \right]$, the natural policy gradient method updates as follows:

$$\theta_{t+1} = \theta_t - \beta_t F(\theta_t)^\dagger Q_{\theta_t}(s_t, a_t) \nabla_\theta \log \pi_{\theta_t}(a_t \mid s_t),$$

where $F(\theta_t)^\dagger$ denotes the Moore-Penrose pseudoinverse of $F(\theta_t)$.

In practice, $F(\theta_t)$ and $Q_{\theta_t}(s_t, a_t)$ can be estimated via sampling. Both PG algorithm and NPG algorithm apply the Monte Carlo method to estimate the Q-value, which might suffer from large variance and high sample complexity. This motivates us to apply AC and NAC algorithms.

We consider the single-loop single-timescale NAC method, where the critic is bootstrapping and uses a single sample to update. The state-value function is approximated by the following linear function

$$Q_\theta(s, a) \approx \phi(s, a)^T \omega_\theta \,,$$

where $\phi(s, a) \in \mathbb{R}^d$ is a known feature mapping for state action pair $(s, a)$, $\omega_\theta \in \Omega \subseteq \mathbb{R}^d$ and we assume $\Omega$ is a compact convex set.

It is usually assumed that the best weight $\omega_\theta$ is unknown while a method to compute the features is available. To compute good weights, the critic will perform a TD(0) update by generating the tuple $(s_t, a_t, r_t, s'_t, a'_t)$ by following the MDP and the policy $\pi_\theta$ and the critic finally performs the update:

$$\omega_{t+1} = P_\Omega \left[ \omega_t + \alpha_t (r_t + \gamma \phi(s'_t, a'_t)^T \omega_t - \phi(s_t, a_t)^T \omega_t) \phi(s_t, a_t) \right] \,,$$

where $\alpha_t \in (0, 1)$ is the step size of the critic's update.

Then the actor can update using the Q-value estimated by the critic. We also need to estimate the fisher information matrix $F(\theta_t)$ via only one sample for the natural actor-critic method. We update its estimation $F_t$ at each iteration $t$ as

$$F_t = (1 - \zeta_t) F_{t-1} + \zeta_t \nabla_\theta \log \pi_{\theta_t}(a_t | s_t) \nabla_\theta \log \pi_{\theta_t}(a_t | s_t)^T \,,$$

where $\zeta_t \in (0, 1)$ is the step size of the Fisher information matrix's update.

**Remark 1.** *In the double-loop NAC algorithm (Xu et al., 2020), $F(\theta_t)$ can be approximated by a sufficient number of samples induced from $\mu_\theta$ in the inner loop, while only one sample can be accessed in the single-loop algorithm. To handle the problem of insufficient samples, we update the estimated Fisher information matrix $F_t$ using $F_{t-1}$ updated in the last iteration. Intuitively, when the policy's distribution shift from $\pi_{\theta_{t-1}}$ to $\pi_{\theta_t}$ is small enough, then $F(\theta_{t-1})$ and $F(\theta_t)$ are also close. Thus it is natural and reasonable to utilize the Fisher information matrix estimated in the previous iterations.*

Thus the actor can update as follows:

$$\theta_{t+1} = \theta_t - \beta_t (F_t + \lambda I)^{-1} \phi(s_t, a_t)^T \omega_t \nabla_\theta \log \pi_{\theta_t}(a_t \mid s_t) \,,$$

where $\lambda I$ is the regularization term to prevent the matrix from being singular, $\beta_t \in (0, 1)$ is the step size.

---

**Algorithm 1** Single-loop single-timescale natural actor critic

1: Initialize at arbitrary $\theta_1, \omega_1, F_0$.
2: **for** time $t = 1, 2, \cdots$ **do**
3:      Generate $(s_t, a_t)$ from $\mu_{\pi_\theta}$, then sample $s'_t \sim P(\cdot \mid s_t, a_t), a'_t \sim \pi_{\theta_t}(\cdot \mid s'_t)$.
4:      Estimated Fisher information matrix update:

$$F_t = (1 - \zeta_t) F_{t-1} + \zeta_t \nabla_\theta \log \pi_{\theta_t}(a_t | s_t) \nabla_\theta \log \pi_{\theta_t}(a_t | s_t)^T \,.$$

5:      Actor update:

$$\theta_{t+1} = \theta_t - \beta_t (F_t + \lambda I)^{-1} \phi(s_t, a_t)^T \omega_t \nabla_\theta \log \pi_{\theta_t}(a_t \mid s_t) \,.$$

6:      Critic update:

$$\omega_{t+1} = P_\Omega \left[ \omega_t + \alpha_t (r_t + \gamma \phi(s'_t, a'_t)^T \omega_t - \phi(s_t, a_t)^T \omega_t) \phi(s_t, a_t) \right] \,.$$

7: **end for**

---

**Remark 2.** *The "single-loop" refers to only one sample being used to update the critic per actor step. The algorithm samples $(s_t, a_t)$ from the stationary distribution $\mu_\theta$ induced by the policy $\pi_{\theta_t}$, which is a mild requirement in the analysis of uniformly ergodic Markov chain. Additionally, the i.i.d. sample is commonly used in the literature of the single-timescale AC analysis (Olshevsky & Gharesifard, 2022; Chen et al., 2021; 2022). Note that many existing theoretical works all start with i.i.d samples from the stationary distribution. It is widely recognized as the first important step toward the analysis of more practical algorithms. In practice, one can run the Markov chain in the simulator a sufficient number of steps and sample one state from the last step. In addition, "single-timescale" refers to the fact that the stepsizes for the critic and the actor updates are constantly proportional.*

## 4 MAIN RESULTS

In this section, we show the finite-time global convergence of the single-loop single-timescale NAC algorithm.

### 4.1 ASSUMPTIONS

Before giving the finite-time convergence guarantee for the single-loop single-timescale NAC algorithm, we first present several standard assumptions commonly used in the literature of analyzing AC (NAC) with linear function approximation.

**Assumption 1** (Uniform mixing). *Let $\mu_{t,\theta}$ be the distribution over state-action pairs after $k$ transitions following policy $\pi_\theta$ in the given MDP. There is a distribution $\mu_\theta$ and constants $C$ and $\rho$ such that*

$$\|\mu_{k,\theta} - \mu_\theta\|_1 \leq C\rho^k$$

This assumption guarantees the existence of the stationary distribution. It is commonly employed to address the issues of the noise induced by Markovian sampling and is widely used in the finite-time analysis of various RL algorithms with Markovian samples (Qiu et al., 2021; Wu et al., 2020; Xu et al., 2020; Olshevsky & Gharesifard, 2022; Chen et al., 2021; Chen & Zhao, 2022).

**Assumption 2** (Smoothness of policy). *The function $\pi_\theta(a|s)$ is a twice continuously differentiable function of $\theta$ for all state-action pairs $s, a$. Further, there exist constants $K_1, K_2, K_3$ such that for all $\theta, s, a$ we have*

$$|\nabla_\theta \log \pi_\theta(a|s)| \leq K_1, |\nabla_\theta \pi_\theta(a|s)| \leq K_2, |\nabla_\theta^2 \log \pi_\theta(a|s)| \leq K_3.$$

This assumption is also standard in the literature of policy gradient methods (Qiu et al., 2021; Wu et al., 2020; Xu et al., 2020; Olshevsky & Gharesifard, 2022; Chen et al., 2021; Chen & Zhao, 2022). It holds for many policy classes, such as tabular softmax policy and Gaussian policy. This assumption ensures that the quantities throughout the execution of NAC are smooth.

**Assumption 3** (Nonredundancy and norm of features). *The feature matrix $\Phi$ is non-singular and each of its rows has at most unit norm.*

This assumption states that the features are not redundant, ensuring the critic has the unique optimal solution when approximating the value function. It is standard and assumed in previous work (Olshevsky & Gharesifard, 2022).

**Assumption 4** (Approximation of the Q-value). *Define $\omega_\theta$ to be the limit of temporal difference update when the policy is fixed to $\pi_\theta$. Then for some $\delta > 0$,*

$$\sup_\theta \mathbb{E}_{s,a\sim\mu_\theta} \left| Q_\theta(s,a) - \phi(s,a)^T \omega_\theta \right| \leq \delta.$$

This assumption is standard in the literature of linear function approximation settings (Qiu et al., 2021; Wu et al., 2020; Xu et al., 2020; Olshevsky & Gharesifard, 2022; Chen et al., 2021; Chen & Zhao, 2022). Here $\delta$ determines how well the Q-value is approximated and $\delta = 0$ when the linear approximation perfectly describes the Q-value functions.

Before making the next assumption, we introduce some notation. Rewrite the critic update as

$$\omega_{t+1} = P_\Omega \left[ (I + \alpha_t A_t)\omega_t - \alpha_t b_t \right], \tag{1}$$

where $A_t, b_t$ are defined as

$$A_t = \phi(s_t, a_t)(\gamma\phi(s_t', a_t') - \phi(s_t, a_t))^T, b_t = -r_t \phi(s_t, a_t).$$

**Assumption 5** (Exploration). *Denote $A_{\theta_t} = \mathbb{E}[A_t]$ and $b_{\theta_t} = \mathbb{E}[b_t]$ where the expectation is being taken by generating the sample from $\mu_{\theta_t}$. There exists $\mu \in (0,1)$ such that*

$$\sup_\theta \sup_{\|x\|=1} x^T A_\theta x \leq -\frac{\mu}{2} < 0$$

The assumption is labeled "exploration" because it holds if the policies $\pi_\theta$ explore all state-action pairs. It is standard in the literature of TD learning with linear function approximation. In the same way, the projected Bellman error is strongly convex, it is known that A is positive definite. Such an assumption is made to guarantee the problem is solvable Kumar et al. (2023); Qiu et al. (2021); Olshevsky & Gharesifard (2022); Chen et al. (2021); Chen & Zhao (2022).

## 4.2 FINITE-TIME CONVERGENCE

For convenience, we use the shorthand $\Delta_t = \omega_t - \omega_{\theta_t}$ to denote the critic estimated error, and denote $\nabla_t = \nabla V(\theta_t)$ as the actor stationary. Further, we use the shorthand $e_t = F_t - F(\theta_t)$ as the Fisher information estimated error. We show that these three errors converge to zero on average. It is convenient to take the average over the last half of the iteration, which is a slight modification.

**Theorem 1.** *Suppose Assumptions 1 - 5 hold. By choosing step sizes $\alpha_t = c_1/\sqrt{t}$, $\beta_t = c_2/\sqrt{t}$, $\zeta_t = c_3/\sqrt{t}$, where $c_1, c_2, c_3$ are appropriate constants chosen depending on the problem parameters, the sequence of iterates produced by single-loop single-timescale NAC satisfies*

$$\frac{2}{T} \sum_{t=T/2}^{T-1} \mathbb{E}\left[\|e_t\|^2\right] \leq O\left(\delta^2 + \frac{1}{\sqrt{T}}\right),$$

$$\frac{2}{T} \sum_{t=T/2}^{T-1} \mathbb{E}\left[\|\nabla_t\|^2\right] \leq O\left(\delta^2 + \frac{1}{\sqrt{T}}\right),$$

$$\frac{2}{T} \sum_{t=T/2}^{T-1} \mathbb{E}\left[\|\Delta_t\|^2\right] \leq O\left(\delta^2 + \frac{1}{\sqrt{T}}\right),$$

*where all parameters except $\delta, T$ are treated as constants in the $O(\cdot)$ notation.*

The above results show that if the critic approximation error $\delta = 0$, the Fisher information estimator, the critic, and the actor all converge at a sub-linear rate of $O(T^{-1/2})$. Note that $O(T^{-1/2})$ is the rate one would obtain from stochastic gradient descent (SGD) on a non-convex function with unbiased gradient updates. In terms of sample complexity, to obtain an $\epsilon$-approximate stationary point, it takes a number of $O(\epsilon^{-2})$ samples, which matches the state-of-the-art performance of SGD on the non-convex optimization problem. This result also matches the state-of-the-art result of single-timescale AC (Olshevsky & Gharesifard, 2022; Chen & Zhao, 2022; Chen et al., 2021).

Given the finite-time convergence of $\Delta_t, \nabla_t, e_t$, we can further attain the globally optimal solution in terms of the V-value function convergence. Note that this is due to the parameter invariant property of the NPG update. We define $\delta' = \max_{\theta \in \Theta} \min_p \left|\mathbb{E}_{\mu_\theta}\left[\nabla_\theta \log \pi_\theta(a|s)^T p - A_{\pi_\theta}(s,a)\right]\right|$, where $A_{\pi_\theta}(s,a) = V_\theta(s) - Q_\theta(s,a)$ is the advantage value function corresponds to $\pi_\theta$. $\delta'$ is the approximation error caused by the insufficient expressive power of the parameterized policy class $\Theta$. It can be shown that $\delta'$ is zero or very small when the policy class is sufficient to express all possible policies, such as the tabular policy or over-parameterized neural policy (Wang et al., 2019). This error definition is also used in previous work when analyzing the global convergence of the NAC (Xu et al., 2020).

**Theorem 2.** *Suppose Assumptions 1 - 5 hold. By choosing step sizes $\alpha_t = c_1/\sqrt{t}$, $\beta_t = c_2/\sqrt{t}$, $\zeta_t = c_3/\sqrt{t}$, where $c_1, c_2, c_3$ are appropriate constants chosen depending on the problem parameters, the value functions $V(\theta_t)$ produced by single-loop single-timescale NAC satisfies*

$$V^* - \frac{2}{T} \sum_{t=T/2}^{T-1} \mathbb{E}\left[V(\theta_t)\right] \leq O\left(\delta^2 + \delta' + T^{-1/4}\right),$$

*where all parameters except $\delta', \delta, T$ are treated as constants in the $O(\cdot)$ notation.*

The above result shows that if the critic approximation error $\delta = 0$ and the actor approximation error $\delta' = 0$, then the value function $V(\theta_t)$ will converge to the globally optimal value function $V^*$ at a sub-linear rate $O(T^{-1/4})$. Thus we can infer that it takes $O(\epsilon^{-4})$ samples to obtain the $\epsilon$-approximate globally optimal point. It is essential to note that we are the first to provide the finite time global convergence for the single-loop single-timescale NAC methods with linear function approximation.

The "two-timescale" step size is primarily considered for the ease of establishing convergence, wherein the analysis heavily relies on the favorable property of $\lim_{t\to\infty} \alpha_t/\beta_t = 0$. Unlike the "two-timescale" approach, the single-timescale step size considered in our paper does not possess

such a property, and our proof does not require it either. It is true that, in practice, we often select an actor's step size smaller than the critic's step size, but they always shrink at the same scale. This is also the single-timescale setting, while the constants $c_1, c_2$ we choose in Theorem 1.

## 5 PROOF SKETCH

This section gives a proof sketch for the finite time convergence results of the single-loop single-timescale NAC algorithm. The detailed proof can be found in Appendix A.

The main challenge in the finite-time analysis for single-loop single-timescale NAC algorithm is that the estimation errors of the Fisher information matrix, critic, and the policy gradient are strongly coupled. To control these three errors simultaneously, we view the propagation of these errors as an interconnected system.

### 5.1 ANALYSIS OF CRITIC UPDATE

Our first step is to obtain a performance error bound on the critic. The final bound we will derive in this subsection will bound the critic's performance $\|\Delta_t\|$ in terms of the closeness to the stationary point of the actor, i.e., $\|\nabla_t\|$.

We first show that, without noise, a small enough TD learning step starting from $\omega_t$ makes it closer to $\omega_{\theta_t}$. Recall that the critic update rule can be rewritten as in Eq. (1). We then denote $\tilde{\nabla}(\theta_t, \omega_t) := -A_{\theta_t}\omega_t + b_{\theta_t}$, where $A_{\theta_t}$ and $b_{\theta_t}$ are defined in Assumption 5.

**Lemma 1** (Contractivity of the TD update). *If $\alpha_t \leq \mu/(2L_\nabla^2)$, then*

$$\left\| \omega_t - \omega_{\theta_t} - \alpha_t \tilde{\nabla}(\theta_t, \omega_t) \right\| \leq (1 - \alpha_t \mu/4) \|\Delta_t\| .$$

The definition of $L_\nabla$ can be found in Appendix A. The above lemma states the fact that following the expected TD direction can reduce the estimated error $\|\Delta_t\|$. With this lemma, we can derive a recursion relation of the critic update.

**Lemma 2** (Recursion relation of the critic update). *If $\alpha_t \leq \mu/(2L_\nabla^2)$, then*

$$
\begin{aligned}
\mathbb{E}\left[ \left\| \omega_{t+1} - \omega_{\theta_{t+1}} \right\|^2 \right] \leq & (1 - \alpha_t \mu/4) \left\| \omega_t - \omega_{\theta_t} \right\|^2 + \alpha_t^2 \sigma_c^2 \\
& + 2L_\omega^2 \beta_t^2 \lambda^{-2}(\sigma_a^2 + 3L_g^2 \mathbb{E}\left[ \|\Delta_t\|^2 \right] + 3\mathbb{E}\left[ \|\nabla_t\|^2 \right] + 3\bar{\delta}^2) \\
& + \beta_t \lambda^{-1} L_\omega \frac{\mathbb{E}\left[ \|\nabla_t\|^2 \right]}{2} + \beta_t \lambda^{-1}(1/2 + L_g)\mathbb{E}\left[ \|\Delta_t\|^2 \right] \\
& + \left( \beta_t^2 \lambda^{-2} \frac{\lambda'}{2} \sigma_{a'}^2 + \beta_t \lambda^{-1} L_\omega \bar{\delta} \right) \|\Delta_t\| ,
\end{aligned}
$$

*where $\sigma_c, L_\omega, \sigma_a, L_g, \bar{\delta}, \sigma_{a'}', \lambda'$ are all problem parameters and are treated as constants. Their definition can be found in the Appendix A.*

With the above recursion relation of the critic update, we can take summation over $t = T/2, \cdots, T-1$ and obtain the bound of $\sum_{t=T/2}^{T-1} \mathbb{E}\left[ \Delta_t^2 \right]$ in terms of $\sum_{t=T/2}^{T-1} \mathbb{E}\left[ \nabla_t^2 \right]$ and other problem parameters.

**Lemma 3.** *Suppose that $\alpha_t, \beta_t$ are non-increasing sequences with $\alpha_{t/2} \leq c_\alpha \alpha_t$ and $\beta_{t/2} \leq c_\beta \beta_t$ for all $t$ and for some constants $c\alpha, c_\beta$. If $\alpha_t \leq \mu/(2L_\nabla^2)$, then*

$$
\begin{aligned}
\frac{2}{T} \sum_{t=T/2}^{T-1} \mathbb{E}\left[ \|\Delta_t\| \right]^2 \leq & \frac{1}{\alpha_T} \frac{8}{\mu T} \left\| \omega_{T/2} - \omega_{\theta_{T/2}} \right\|^2 + c_\alpha^2 \alpha_T \frac{8}{\mu} \sigma_c^2 \\
& + 2L_\omega^2 \frac{c_\beta^2 \beta_T^2 \lambda^{-2}}{\alpha_T} \frac{8}{\mu} \sigma_a^2 + 2L_\omega^2 \frac{c_\beta^2 \beta_T^2 \lambda^{-2}}{\alpha_T} \frac{8}{\mu} \delta^2 \\
& + \frac{4c_\beta^2 \beta_T^2 \sigma_{a'}^2 \lambda' + 8c_\beta \beta_T \lambda^{-1} L_\omega \delta}{\alpha_T \mu} \sqrt{\frac{1}{T/2} \sum_{t=T/2}^{T-1} \mathbb{E}\left[ \|\Delta_t\|^2 \right]}
\end{aligned}
$$

$$+ \frac{8c_\beta \beta_T \lambda^{-1} L_\omega}{\alpha_T \mu} \frac{2}{T} \sum_{t=T/2}^{T-1} \mathbb{E}\left[\|\nabla_t\|^2\right] .$$

## 5.2 ANALYSIS OF ACTOR UPDATE

Now we analyze the actor's stationary $\|\nabla_t\|$. It depends on the critic estimating error $\|\Delta_t\|$ and the fisher information matrix approximating error $\|F_t - F(\theta_t)\|$. Using the actor update rule and the smoothness property of $V(\theta)$, we derive

**Lemma 4.** *Suppose $\beta_t \leq \frac{\lambda^2}{4(\lambda+1)L_V}$ for all $t \geq T/2$, then*

$$\frac{2}{T} \sum_{t=T/2}^{T-1} \mathbb{E}\left[\|\nabla_t\|^2\right] \leq \frac{16\mathbb{E}\left[V(\theta_{T/2})\right] - 16\mathbb{E}\left[V(\theta_T)\right]}{\beta_T T} + \frac{36c_\beta L_g^2}{\lambda^2 T/2} \sum_{t=T/2}^{T-1} \mathbb{E}\left[\|\Delta_t\|^2\right] + \frac{36c_\beta \delta^2}{\lambda^2}$$

$$+ \frac{18c_\beta \beta_T L_V \sigma_a^2}{\lambda^2} + \frac{8c_\beta}{\lambda^2(1-\gamma)^2 T/2} \sum_{t=T/2}^{T-1} \mathbb{E}\left[\|e_t\|^2\right] .$$

It can be seen that the actor stationary $\|\nabla_t\|$ can be bounded by the actor's performance difference, the critic estimated error $\|\Delta_t\|$, the Fisher information matrix estimated error $\|e_t\|$, and other related problem parameters such as the critic's approximation bias $\delta$ and the variance upper bound $\sigma_a$.

## 5.3 ANALYSIS OF THE FISHER INFORMATION ESTIMATOR

For the Fisher information matrix estimated error $\|e_t\|$, recall that $F_{t+1}$ is updated as follows:

$$F_t = (1 - \zeta_t)F_{t-1} + \zeta_t \nabla_\theta \log \pi(a_t|s_t) \nabla_\theta \log \pi(a_t|s_t)^T .$$

Then we can decompose the difference of $F_t$ and $F_{\theta_t}$ as

$$\begin{aligned} F_t - F(\theta_t) =& (1 - \zeta_t)F_{t-1} + \zeta_t \nabla_\theta \log \pi(a_t|s_t) \nabla_\theta \log \pi(a_t|s_t)^T - F(\theta_t) \\ =& (1 - \zeta_t)(F_{t-1} - F(\theta_{t-1})) + \zeta_t(\nabla_\theta \log \pi(a_t|s_t) \nabla_\theta \log \pi(a_t|s_t)^T - F(\theta_t)) \\ & + (1 - \zeta_t)(F(\theta_{t-1}) - F(\theta_t)) . \end{aligned}$$

This leads to a contractive property similar to the critic update. It can be seen that $\|e_t\|$ is controlled by the $\|e_{t-1}\|$ in the last iteration, the variance of the unbiased estimator using the current sample, and the difference between $F(\theta_t)$ and $F(\theta_{t-1})$, which is related to $\|\nabla_t\|$ and $\|\Delta_t\|$.

**Lemma 5.** *Suppose that $\zeta_t$ are non-increasing sequences satisfying $\zeta_{t/2} \leq c_\zeta \zeta_t$ and for all $t$ and for constant $c_\zeta$. For the Fisher information matrix estimator, we have*

$$\frac{2}{T} \sum_{t=T/2}^{T-1} \mathbb{E}\left[\|e_t\|^2\right] \leq \frac{1}{\zeta_T T} \left\|F_{T/2} - F(\theta_{T/2})\right\|^2 + c_\zeta^2 \zeta_T \sigma_F^2$$

$$+ 2L_F^2 \frac{c_\beta^2 \beta_T^2 \lambda^{-2}}{\zeta_T} \sigma_a^2 + 2L_F^2 \frac{c_\beta^2 \beta_T^2 \lambda^{-2}}{\zeta_T} \bar{\delta}^2$$

$$+ \frac{c_\beta \beta_T \lambda^{-1} L_F}{\zeta_T} \frac{2}{T} \sum_{t=T/2}^{T-1} \mathbb{E}\left[\|\nabla_t\|\right]^2$$

$$+ \frac{c_\beta \beta_T \lambda^{-1} L_g L_F}{\zeta_T} \frac{2}{T} \sum_{t=T/2}^{T-1} \mathbb{E}\left[\|\Delta_t\|\right]^2 ,$$

*where $L_F, \sigma_F$ is the problem parameter, and its definition can be found in Appendix A.*

## 5.4 INTERCONNECTED ITERATION SYSTEM ANALYSIS

Now we have already controlled the approximating errors of the Fisher information matrix, critic, and the policy gradient. We define $X_T$, $Y_T$, and $Z_T$ as the expectation of the critic error, the square

norm of the policy gradient, and the Fisher information estimated error, respectively:

$$X_T = \frac{2}{T} \sum_{t=T/2}^{T-1} \mathbb{E}\left[\|\Delta_t\|^2\right], Y_T = \frac{2}{T} \sum_{t=T/2}^{T-1} \mathbb{E}\left[\|\nabla_t\|^2\right], Z_T = \frac{2}{T} \sum_{t=T/2}^{T-1} \mathbb{E}\left[\|e_t\|^2\right].$$

By choosing $\alpha_t, \beta_t, \gamma_t = O(1/\sqrt{t})$ We can get the following inequalities:

$$X_T \leq O\left(\frac{1}{\sqrt{T}}\right) + C_1 \sqrt{X_T} + C_2 Y_T,$$

$$Y_T \leq O\left(\frac{1}{\sqrt{T}}\right) + C_3 X_T + C_4 Z_T + C_5,$$

$$Z_T \leq O\left(\frac{1}{\sqrt{T}}\right) + C_6 X_T + C_7 Y_T,$$

where $C_1$ to $C_7$ are all positive constants. by solving the above three inequalities, we show that when the above constants satisfy $1 - C_4 C_7 > 0$ and $1 - 2C_2 \frac{C_3 + C_4 C_6}{1 - C_4 C_7} > 0$, $X_T, Y_T, Z_T$ all converge at a rate of $O(1/\sqrt{T} + \delta^2)$. This condition can be satisfied by tuning the step sizes $\alpha_t, \beta_t, \zeta_t$. Thus, it completes the proof of convergence to a stationary point.

Given the convergence results of these three errors, we can further derive the convergence to the globally optimal point, which leverages the parameter invariant property of the NPG update. The algorithm needs $O(\epsilon^{-4})$ number of samples to attain the $\epsilon$-accurate global optimality, i.e., $V^* - V(\theta_t) \leq \epsilon$. Due to the page limit, we defer to Appendix A for detailed proof.

## 6 CONCLUSION

In this paper, we provide the first finite-time sample complexity guarantee for the single-loop single-timescale NAC algorithm, which needs $O\left(\epsilon^{-2}\right)$ samples to find an $\epsilon$-approximate stationary point and $O\left(\epsilon^{-4}\right)$ to find an $\epsilon$-global optimal value function.

The novelty of this work stems from the design of the Fisher information matrix estimator. To decrease the variance of the Fisher information matrix estimation, we combine the previous estimation with the unbiased estimation using a single sample. This update rule links the estimation error to the actor's and critic's errors. Intuitively, when the actor takes a sufficiently small step, the Fisher information matrix estimation is accurate as it does not change substantially between two iterations. In our theoretical analysis, we derive the recursion relation for this estimator and control the error with the critic error and the norm of the policy gradient.

**Future Work** Our work assumes that at each iteration, we can draw a sample from the stationary distribution $\mu_{\theta_t}$, which may not be so easy to obtain in practice. It will be more practical if the sample is Markovian, which means the sample is taken from the state of the last iteration and the action taken by following $\pi_{\theta_t}$. The analysis is more complicated for Markovian samples, and we leave it as an interesting future work.

Note that our result builds on the finite-time analysis, which may not imply the asymptotic convergence. This finite-time objective is commonly used in the optimization literature, and we leave it as a future work to further show the asymptotic guarantee.

Moreover, there still remains a gap between our globally optimal $O(\epsilon^{-4})$ sample complexity and the state-of-the-art $O(\epsilon^{-3})$ complexity obtained when analyzing double-loop NAC method Xu et al. (2020). This is because in single-loop single-timescale NAC, the actor and the critic both use $O(1/\sqrt{t})$ step sizes and thus we are unable to balance the terms in the global optimality analysis. It should be noted that the convergence result of attaining a stationary point matches the state-of-the-art sample complexity when analyzing AC (NAC). Thus our result is still comparable, and we leave it as future work to derive a tighter sample complexity for globally optimal convergence.

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

# A  PROOF OF MAIN RESULTS

## A.1  REFORMULATING NAC AND PROPERTIES OF UPDATES

Rewrite the critic update rule as

$$\omega_{t+1} = P_\Omega \left[ \omega_t - \alpha_t \left( \tilde{\nabla}(\theta_t, \omega_t) + w_c(t) \right) \right] .$$

$w_c(t)$ is a random variable with zero expectation conditioned on the entire history. $\tilde{\nabla}(\theta_t, \omega_t) := -A_{\theta_t}\omega_t + b_{\theta_t}$ is the expected TD direction. Note that $\omega_\theta$ is the limiting point of TD learning. Thus we have

$$\tilde{\nabla}(\theta, \omega_\theta) = 0 .$$

Further, it can be implied that

$$
\begin{aligned}
\tilde{\nabla}(\theta, \omega)^T (\omega - \omega_\theta) &= (-A_\theta \omega + b_\theta)^T (\omega - \omega_\theta) \\
&= (-A_\theta \omega + b_\theta)^T (\omega - \omega_\theta) - (-A_\theta \omega_\theta + b_\theta)^T (\omega - \omega_\theta) \\
&\geq \frac{\mu}{2} \|\omega - \omega_\theta\|^2
\end{aligned}
$$

Then we turn to the actor update. We can rewrite the actor update as

$$\theta_{t+1} = \theta_t - \beta_t (F_t + \lambda I)^{-1} \left( \sum_{s,a} \mu_{\theta_t}(s,a) Q_t(s,a) \nabla_\theta \log \pi_{\theta_t}(a|s) + w_a(t) \right) ,$$

where $w_a(t)$ is a random variable with zero expectation conditional on the past trajectory, $Q_t(s,a) = \phi(s,a)^T \omega_t$.

Then we define $\hat{g}(\theta, Q) = \sum_{s,a} \mu_\theta(s,a) Q(s,a) \nabla_\theta \pi_\theta(a|s)$, and define $g(\theta, \omega) = \hat{g}(\theta, \Phi\omega)$, where $\Phi$ is the feature matrix. Then the actor's update can be rewritten as

$$\theta_{t+1} = \theta_t - \beta_t (F_t + \lambda I)^{-1} \left( g(\theta_t, \omega_t) + w_a(t) \right) .$$

By Assumption 2 we have

$$
\begin{aligned}
\|g(\theta, \omega_\theta) - \hat{g}(\theta, Q_\theta)\| &\leq \sum_{s,a} \left| \phi(s,a)^T \omega_\theta - Q_\theta \right| K_1 \\
&\leq K_1 \delta .
\end{aligned}
$$

To analyze gradient descent, one typically needs some assumptions on the underlying functions. These tend to involve the continuity of various gradients and updates, boundedness and finite variance of the noise.

**Lemma 6** (Lipschitz of critic update (Lemma 5.2 in Olshevsky & Gharesifard (2022))). *There exists a constant $L_\nabla < \infty$ such that for all $\theta, \omega_1, \omega_2$,*

$$\left\| \tilde{\nabla}(\theta, \omega_1) - \tilde{\nabla}(\theta, \omega_2) \right\| \leq L_\nabla \|\omega_1 - \omega_2\| .$$

**Lemma 7** (Lipschitz of actor update (Lemma 5.5 in Olshevsky & Gharesifard (2022))). *There exists a constant $L_g < \infty$. For all $\theta, \omega_1, \omega_2$, we have that*

$$\|g(\theta, \omega_1) - g(\theta, \omega_2)\| \leq L_g \|\omega_1 - \omega_2\| .$$

**Lemma 8** (Lipschitz of the TD fixed point (Lemma 5.10 in Olshevsky & Gharesifard (2022))). *There exists a constant $L_\omega < \infty$ such that*

$$\|\omega_{\theta_1} - \omega_{\theta_2}\| \leq L_\omega \|\theta_1 - \theta_2\| .$$

**Lemma 9** (Lipschitz gradient for value function (Proposition 5.7 in Olshevsky & Gharesifard (2022))). *There exists some $L_V < \infty$ such that the function $V(\theta)$ has $L_V$-Lipschitz gradient*

$$\|\nabla_\theta V(\theta_1) - \nabla_\theta V(\theta_2)\| \leq L_V \|\theta_1 - \theta_2\| .$$

**Lemma 10** (Lipschitz of the Fisher information matrix). *There exists a constant $L_F < \infty$ such that*

$$\|F(\theta_1) - F(\theta_2)\| \le L_F \|\theta_1 - \theta_2\| .$$

*Proof.*

$$
\begin{aligned}
&\|F(\theta_1) - F(\theta_2)\| \\
&= \left\| \mathbb{E}_{\theta_1} \left[ \nabla_\theta \log \pi_{\theta_1}(a|s) \nabla_\theta \log \pi_{\theta_1}(a|s)^T \right] - \mathbb{E}_{\theta_2} \left[ \nabla_\theta \log \pi_{\theta_2}(a|s) \nabla_\theta \log \pi_{\theta_2}(a|s)^T \right] \right\| \\
&\le \max_{s,a} \left\| \nabla_\theta \log \pi_{\theta_1}(a|s) \nabla_\theta \log \pi_{\theta_1}(a|s)^T - \nabla_\theta \log \pi_{\theta_2}(a|s) \nabla_\theta \log \pi_{\theta_2}(a|s)^T \right\| \\
&\le |S||A| K_1^2 .
\end{aligned}
$$

$\square$

**Lemma 11** (Bounded support for critic noise (Lemma 5.3 in Olshevsky & Gharesifard (2022))). *The support of the random vector $w_c(t)$ belongs to some compact set. And then there exist constants $\sigma_c < \infty$ and $\sigma_{c'} < \infty$ such that for all $t$,*

$$\mathbb{E}\left[ \|w_c(t)\|^2 \,|\mathcal{F}_t \right] \le \sigma_c^2$$

$$\sqrt{\mathbb{E}\left[ \|w_c(t)\|^4 \,|\mathcal{F}_t \right]} \le \sigma_{c'}^2 .$$

**Lemma 12** (Bounded support for actor noise (Lemma 5.4 in Olshevsky & Gharesifard (2022))). *The support of the random vector $w_a(t)$ belongs to some compact set. And then there exist constants $\sigma_a < \infty$ and $\sigma_{a'} < \infty$ such that for all $t$,*

$$\mathbb{E}\left[ \|w_a(t)\|^2 \,|\mathcal{F}_t \right] \le \sigma_a^2$$

$$\sqrt{\mathbb{E}\left[ \|w_a(t)\|^4 \,|\mathcal{F}_t \right]} \le \sigma_{a'}^2 .$$

**Lemma 13** (Bounded curvature of the TD fixed point (Lemma 5.11 in Olshevsky & Gharesifard (2022))). *There is some quantity $\lambda_i$ independent of $\theta$ such that*

$$\sup_\theta \lambda_{\max}(\nabla_\theta^2 \omega_\theta(i)) \le \lambda_i .$$

*We can further define $\lambda' = \sqrt{\sum_i \lambda_i^2}$.*

### A.2 ANALYSIS OF CRITIC UPDATE

Our first step is to obtain a performance error bound on the critic. The final bound we will derive in this subsection will bound the critic's performance in terms of the closeness to the stationary point of the actor.

**Lemma 14** (Contractivity of the TD update). *If $\alpha_t \le \mu/(2L_\nabla^2)$, then*

$$\left\| \omega_t - \omega_{\theta_t} - \alpha_t \tilde{\nabla}(\theta_t, \omega_t) \right\| \le (1 - \alpha_t \mu/4) \|\Delta_t\| .$$

*Proof.*

$$
\begin{aligned}
\left\| \omega_t - \omega_{\theta_t} - \alpha_t \tilde{\nabla}(\theta_t, \omega_t) \right\|^2 &= \|\omega_t - \omega_{\theta_t}\|^2 - 2\alpha_t \tilde{\nabla}(\theta_t, \omega_t) + \alpha_t^2 \left\| \tilde{\nabla}(\theta_t, \omega_t) \right\|^2 \\
&\le \|\omega_t - \omega_{\theta_t}\|^2 - 2\alpha_t \frac{\mu}{2} \|\omega_t - \omega_{\theta_t}\|^2 + \alpha_t^2 \left\| \tilde{\nabla}(\theta_t, \omega_t) \right\|^2 \\
&\le (1 - \alpha_t \mu + L_\nabla^2 \alpha_t^2) \|\omega_t - \omega_{\theta_t}\|^2 \\
&\le (1 - \alpha_t \mu/2) \|\omega_t - \omega_{\theta_t}\|^2
\end{aligned}
$$

$\square$

**Proof of Lemma 2**   Recall that the critic update can be rewritten as

$$\omega_{t+1} = \omega_t - \alpha_t \left( \tilde{\nabla}(\theta_t, \omega_t) + w_c(t) \right),$$

where $\tilde{\nabla}(\theta_t, \omega_t) = -A_{\theta_t}\omega_t + b_{\theta_t}$ and $w_c(t) = -(A_t - A_{\theta_t})\omega_t + (b_t - b_{\theta_t})$.

From the critic update rule, we have that

$$\left\| \omega_{t+1} - \omega_{\theta_{t+1}} \right\|^2 = \left\| \omega_t - \alpha_t \tilde{\nabla}(\theta_t, \omega_t) - \alpha_t w_c(t) - \omega_{\theta_{t+1}} \right\|^2$$

$$= \left\| (\omega_t - \omega_{\theta_t} - \alpha_t \tilde{\nabla}(\theta_t, \omega_t) - \alpha_t w_c(t)) + (\omega_{\theta_t} - \omega_{\theta_{t+1}}) \right\|^2$$

Taking the expectation of both sides, we obtain

$$\mathbb{E}\left[ \left\| \omega_{t+1} - \omega_{\theta_{t+1}} \right\|^2 |\mathcal{F}_t \right] \leq \mathbb{E}\left[ \left\| \omega_t - \omega_{\theta_t} - \alpha_t \tilde{\nabla}(\theta_t, \omega_t) - \alpha_t w_c(t) \right\|^2 |\mathcal{F}_t \right]$$

$$+ \mathbb{E}\left[ \left\| \omega_{\theta_t} - \omega_{\theta_{t+1}} \right\|^2 |\mathcal{F}_t \right]$$

$$+ 2\mathbb{E}\left[ \left( \omega_{\theta_t} - \omega_{\theta_{t+1}} \right)^T \left( \omega_t - \omega_{\theta_t} - \alpha_t \tilde{\nabla}(\theta_t, \omega_t) - \alpha_t w_c(t) \right) |\mathcal{F}_t \right]$$

$$\leq \mathbb{E}\left[ \left\| \omega_t - \omega_{\theta_t} - \alpha_t \tilde{\nabla}(\theta_t, \omega_t) \right\|^2 |\mathcal{F}_t \right] + \mathbb{E}\left[ \left\| \alpha_t w_c(t) \right\|^2 |\mathcal{F}_t \right]$$

$$- 2\langle \omega_t - \omega_{\theta_t} - \alpha_t \tilde{\nabla}(\theta_t, \omega_t), \alpha_t w_c(t) \rangle$$

$$+ \mathbb{E}\left[ \left\| \omega_{\theta_t} - \omega_{\theta_{t+1}} \right\|^2 |\mathcal{F}_t \right]$$

$$+ 2\mathbb{E}\left[ \left( \omega_{\theta_t} - \omega_{\theta_{t+1}} \right)^T \left( \omega_t - \omega_{\theta_t} - \alpha_t \tilde{\nabla}(\theta_t, \omega_t) - \alpha_t w_c(t) \right) |\mathcal{F}_t \right]$$

$$\leq (1 - \alpha_t \mu/4) \left\| \omega_t - \omega_{\theta_t} \right\|^2 + \alpha_t^2 \sigma_c^2$$

$$+ \mathbb{E}\left[ \left\| \omega_{\theta_t} - \omega_{\theta_{t+1}} \right\|^2 |\mathcal{F}_t \right]$$

$$+ 2\mathbb{E}\left[ \left( \omega_{\theta_t} - \omega_{\theta_{t+1}} \right)^T \left( \omega_t - \omega_{\theta_t} - \alpha_t \tilde{\nabla}(\theta_t, \omega_t) - \alpha_t w_c(t) \right) |\mathcal{F}_t \right]$$

$$\leq (1 - \alpha_t \mu/4) \left\| \omega_t - \omega_{\theta_t} \right\|^2 + \alpha_t^2 \sigma_c^2$$

$$+ 2L_\omega^2 \beta_t^2 \lambda^{-2}(\sigma_a^2 + \| g(\theta_t, \omega_t) \|^2)$$

$$+ \beta_t \lambda^{-1} L_\omega \| g(\theta_t, \omega_t) \| \left\| \omega_t - \omega_{\theta_t} - \alpha_t \tilde{\nabla}(\theta_t, \omega_t) \right\|$$

$$+ \beta_t^2 \lambda^{-2} \frac{\lambda'}{2} \sigma_{a'}^2 \| \Delta_t \|$$

$$\leq (1 - \alpha_t \mu/4) \left\| \omega_t - \omega_{\theta_t} \right\|^2 + \alpha_t^2 \sigma_c^2$$

$$+ 2L_\omega^2 \beta_t^2 \lambda^{-2}(\sigma_a^2 + \| g(\theta_t, \omega_t) \|^2)$$

$$+ \beta_t \lambda^{-1} L_\omega \| g(\theta_t, \omega_t) \| \| \Delta_t \|$$

$$+ \beta_t^2 \lambda^{-2} \frac{\lambda'}{2} \sigma_{a'}^2 \| \Delta_t \|.$$

For term $\mathbb{E}\left[ \left\| \omega_{\theta_t} - \omega_{\theta_{t+1}} \right\|^2 |\mathcal{F}_t \right]$, we have

$$\mathbb{E}\left[ \left\| \omega_{\theta_t} - \omega_{\theta_{t+1}} \right\|^2 |\mathcal{F}_t \right] \leq L_\omega^2 \mathbb{E}\left[ \left\| \theta_{t+1} - \theta_t \right\|^2 |\mathcal{F}_t \right]$$

$$\leq L_\omega^2 \mathbb{E}\left[ \left\| \beta_t (F_t + \lambda I)^{-1} (g(\theta_t, \omega_t) + w_a(t)) \right\|^2 |\mathcal{F}_t \right]$$

$$\leq L_\omega^2 \beta_t^2 \lambda^{-2} \mathbb{E}\left[ \left\| (g(\theta_t, \omega_t) + w_a(t)) \right\|^2 |\mathcal{F}_t \right]$$

$$\leq 2L_\omega^2 \beta_t^2 \lambda^{-2} (\mathbb{E}\left[ \left\| (g(\theta_t, \omega_t) \right\|^2 |\mathcal{F}_t \right] + \sigma_a^2).$$

For the term $\mathbb{E}\left[\left(\omega_{\theta_t} - \omega_{\theta_{t+1}}\right)^T \left(\omega_t - \omega_{\theta_t} - \alpha_t \tilde{\nabla}(\theta_t, \omega_t) - \alpha_t w_c(t)\right) | \mathcal{F}_t\right]$, we further define $\theta_{t+1/2} := \theta_t - \beta_t g(\theta_t, \omega_t)$ and thus $\theta_{t+1} = \theta_{t+1/2} - \beta_t w_a(t)$. Then we have

$$\mathbb{E}\left[\left(\omega_{\theta_t} - \omega_{\theta_{t+1}}\right)^T \left(\omega_t - \omega_{\theta_t} - \alpha_t \tilde{\nabla}(\theta_t, \omega_t) - \alpha_t w_c(t)\right) | \mathcal{F}_t\right]$$

$$=\mathbb{E}\left[\left(\omega_{\theta_t} - \omega_{\theta_{t+1/2}}\right)^T \left(\omega_t - \omega_{\theta_t} - \alpha_t \tilde{\nabla}(\theta_t, \omega_t) - \alpha_t w_c(t)\right) | \mathcal{F}_t\right]$$

$$+ \mathbb{E}\left[\left(\omega_{\theta_{t+1/2}} - \omega_{\theta_{t+1/2}}\right)^T \left(\omega_t - \omega_{\theta_t} - \alpha_t \tilde{\nabla}(\theta_t, \omega_t) - \alpha_t w_c(t)\right) | \mathcal{F}_t\right]$$

$$\leq L_\omega \mathbb{E}\left[\|g(\theta_t, \omega_t)\| | \mathcal{F}_t\right] \mathbb{E}\left[\left\|\omega_t - \omega_{\theta_t} - \alpha_t \tilde{\nabla}(\theta_t, \omega_t)\right\| | \mathcal{F}_t\right]$$

$$+ \mathbb{E}\left[\left(\omega_{\theta_{t+1/2}} - \omega_{\theta_{t+1/2}}\right)^T \left(\omega_t - \omega_{\theta_t} - \alpha_t \tilde{\nabla}(\theta_t, \omega_t) - \alpha_t w_c(t)\right) | \mathcal{F}_t\right]$$

$$\leq L_\omega \mathbb{E}\left[\|g(\theta_t, \omega_t)\| | \mathcal{F}_t\right] \|\Delta_t\|$$

$$+ \mathbb{E}\left[\left(\omega_{\theta_{t+1/2}} - \omega_{\theta_{t+1/2}}\right)^T \left(\omega_t - \omega_{\theta_t} - \alpha_t \tilde{\nabla}(\theta_t, \omega_t) - \alpha_t w_c(t)\right) | \mathcal{F}_t\right].$$

For the term $\|g(\theta_t, \omega_t)\|$, we have

$$\|g(\theta_t, \omega_t)\| = \|g(\theta_t, \omega_t) + g(\theta_t, \omega_{\theta_t}) - g(\theta_t, \omega_{\theta_t})\|$$

$$\leq L_g \|\omega_t - \omega_{\theta_t}\| + \|g(\theta_t, \omega_{\theta_t})\|$$

$$= L_g \|\Delta_t\| + \|g(\theta_t, \omega_{\theta_t}) - g(\theta_t, Q_{\theta_t}) + g(\theta_t, Q_{\theta_t})\|$$

$$\leq L_g \|\Delta_t\| + \|\nabla_t\| + \bar{\delta}.$$

Using $(a + b + c)^2 \leq 3(a^2 + b^2 + c^2)$, we further get

$$\|g(\theta_t, \omega_t)\|^2 \leq 3L_g^2 \|\Delta_t\|^2 + 3 \|\nabla_t\|^2 + 3\bar{\delta}^2.$$

For the term $\mathbb{E}\left[\left(\omega_{\theta_{t+1/2}} - \omega_{\theta_{t+1/2}}\right)^T \left(\omega_t - \omega_{\theta_t} - \alpha_t \tilde{\nabla}(\theta_t, \omega_t) - \alpha_t w_c(t)\right) | \mathcal{F}_t\right]$, from lemma 5.13 in Olshevsky & Gharesifard (2022), we have that

$$\mathbb{E}\left[\left(\omega_{\theta_{t+1/2}} - \omega_{\theta_{t+1/2}}\right)^T \left(\omega_t - \omega_{\theta_t} - \alpha_t \tilde{\nabla}(\theta_t, \omega_t) - \alpha_t w_c(t)\right) | \mathcal{F}_t\right] \leq \beta_t^2 \lambda' \sigma_{a'}^2 \|\Delta_t\|.$$

Thus we can conclude the proof.

**Proof of Lemma 3** If we have the recursion form

$$x_{t+1} \leq (1 - \zeta)x_t + \epsilon_t,$$

for $0 < \zeta < 1$, then we have

$$\sum_{t=a}^{b} x_t \leq \frac{x_a}{\zeta} + \sum_{t=a}^{b} \frac{\epsilon_t}{\zeta}.$$

Applying this with $\zeta = \alpha_T \mu/4$ since sequence $a_t$ is non-increasing and $\alpha_t \mu \leq 1$ by assumption.

$$\sum_{t=T/2}^{T-1} E\left[\|\omega_t - \omega_{\theta_t}\|\right]^2 \leq \frac{1}{\alpha_T}\frac{4}{\mu} \left\|\omega_{T/2} - \omega_{\theta_{T/2}}\right\|^2 + c_\alpha^2 \alpha_T \frac{T}{2}\frac{4}{\mu}\sigma_c^2$$

$$+ 2L_\omega^2 \frac{c_\beta^2 \beta_T^2}{\alpha_T}\frac{T}{2}\frac{4}{\mu}\sigma_a^2 + 6L_\omega^2 \frac{c_\beta^2 \beta_T^2 \lambda^{-2}}{\alpha_T}\frac{8}{\mu} \sum_{t=T/2}^{T-1} E\|\nabla_t\|^2 + 6L_\omega^2 \frac{c_\beta^2 \beta_T^2 \lambda^{-2}}{\alpha_T}\frac{T}{2}\frac{4}{\mu}\bar{\delta}^2$$

$$+ 6L_\omega^2 \frac{c_\beta^2 \beta_T^2 \lambda^{-2}}{\alpha_T}\frac{8}{\mu}L_g^2 \sum_{t=T/2}^{T-1} E\|\Delta_t\|^2$$

$$+ 4 \frac{c_\beta^2 \beta_T^2 \lambda^{-2} \sigma_{a'}^2 (\lambda'/2) + c_\beta \beta_T L_\omega \bar{\delta}}{\alpha_T \mu} \sqrt{T/2} \sqrt{E \sum_{t=T/2}^{T-1} \|\Delta_t\|^2}$$

$$+ \frac{c_\beta \beta_T \lambda^{-1}}{\alpha_T} \frac{2L_\omega}{\mu} \sum_{t=T/2}^{T-1} E \|\nabla_t\|^2 + \frac{c_\beta \beta_T \lambda^{-1}}{\alpha_T} \frac{2L_\omega}{\mu} \sum_{t=T/2}^{T-1} E \|\Delta_t\|^2$$

$$+ \frac{c_\beta \beta_T \lambda^{-1}}{\alpha_T} \frac{4}{\mu} L_g L_\omega \sum_{t=T/2}^{T-1} E \|\Delta_t\|_2^2$$

Let us observe that the coefficient of $\sum_{t=T/2}^{T-1} \|\Delta_t\|^2$ on the right-hand side is

$$6 L_\omega^2 \frac{c_\beta^2 \beta_T^2 \lambda^{-2}}{\alpha_T} \frac{8}{\mu} L_g^2 + \frac{c_\beta \beta_T \lambda^{-1}}{\alpha_T} \frac{2L_\omega}{\mu} + \frac{c_\beta \beta_T \lambda^{-1}}{\alpha_T} \frac{4}{\mu} L_\omega L_g \leq \frac{1}{2}$$

Then we can have

$$\frac{2}{T} \sum_{t=T/2}^{T-1} \mathbb{E}\left[\|\Delta_t\|\right]^2 \leq \frac{1}{\alpha_T} \frac{8}{\mu T} \left\|\omega_{T/2} - \omega_{\theta_{T/2}}\right\|^2 + c_\alpha^2 \alpha_T \frac{8}{\mu} \sigma_c^2$$

$$+ 2L_\omega^2 \frac{c_\beta^2 \beta_T^2 \lambda^{-2}}{\alpha_T} \frac{8}{\mu} \sigma_a^2 + 2L_\omega^2 \frac{c_\beta^2 \beta_T^2 \lambda^{-2}}{\alpha_T} \frac{8}{\mu} \delta^2$$

$$+ \frac{4c_\beta^2 \beta_T^2 \sigma_{a'}^2 \lambda' + 8 c_\beta \beta_T \lambda^{-1} L_\omega \delta}{\alpha_T \mu} \sqrt{\frac{1}{T/2} \sum_{t=T/2}^{T-1} \mathbb{E}\left[\|\Delta_t\|\right]^2}$$

$$+ \frac{8 c_\beta \beta_T \lambda^{-1} L_\omega}{\alpha_T \mu} \frac{2}{T} \sum_{t=T/2}^{T-1} \mathbb{E}\left[\|\nabla_t\|\right]^2 .$$

### A.3  ANALYSIS OF ACTOR UPDATE

**Proof of Lemma 4**  Now we analyze the actor stationary $\|\nabla_t\|$, it depends on the critic estimated error $\|\Delta_t\|$ and the fisher information matrix estimated error $\|F_t - F(\theta_t)\|$.

Denote $u_t(\omega_t) = (F_t + \lambda I)^{-1} \phi(s_t, a_t)^T \omega_t \nabla_\theta \log \pi_{\theta_t}(a_t \mid s_t)$

$$\mathbb{E}\left[V(\theta_{t+1})\right] \geq \mathbb{E}\left[V(\theta_t)\right] + \mathbb{E}\left[\langle \nabla_\theta V(\theta_t), \theta_{t+1} - \theta_t \rangle\right] - \frac{L_V}{2} \mathbb{E}\left[\|\theta_{t+1} - \theta_t\|^2\right]$$

$$= \mathbb{E}\left[V(\theta_t)\right] + \beta_t \mathbb{E}\left[\langle \nabla_\theta V(\theta_t), u_t(\omega_t) \rangle\right] - \frac{L_V \beta_t^2}{2} \mathbb{E}\left[\|u_t(\omega_t)\|^2\right]$$

$$= \mathbb{E}\left[V(\theta_t)\right] + \beta_t \mathbb{E}\left[\langle \nabla_\theta V(\theta_t), (F(\theta_t) + \lambda I)^{-1} \nabla_\theta V(\theta_t) \rangle\right]$$

$$+ \beta_t \mathbb{E}\left[\langle \nabla_\theta V(\theta_t), u_t(\omega_t) - (F(\theta_t) + \lambda I)^{-1} \nabla_\theta V(\theta_t) \rangle\right]$$

$$- \frac{L_V \beta_t^2}{2} \mathbb{E}\left[\left\|u_t(\omega_t) - (F(\theta_t) + \lambda I)^{-1} \nabla_\theta V(\theta_t) + (F(\theta_t) + \lambda I)^{-1} \nabla_\theta V(\theta_t)\right\|^2\right]$$

$$\geq \mathbb{E}\left[V(\theta_t)\right] + \frac{\beta_t}{1 + \lambda} \mathbb{E}\left[\|\nabla_\theta V(\theta_t)\|^2\right] + \beta_t \mathbb{E}\left[\langle \nabla_\theta V(\theta_t), u_t(\omega_t) - (F(\theta_t) + \lambda I)^{-1} \nabla_\theta V(\theta_t) \rangle\right]$$

$$- L_V \beta_t^2 \mathbb{E}\left[\left\|u_t(\omega_t) - (F(\theta_t) + \lambda I)^{-1} \nabla_\theta V(\theta_t)\right\|^2\right] - L_V \beta_t^2 \mathbb{E}\left[\left\|(F(\theta_t) + \lambda I)^{-1} \nabla_\theta V(\theta_t)\right\|^2\right]$$

$$\geq \mathbb{E}\left[V(\theta_t)\right] + \frac{\beta_t}{1 + \lambda} \mathbb{E}\left[\|\nabla_\theta V(\theta_t)\|^2\right]$$

$$- \beta_t \left(\frac{1}{2(1 + \lambda)} \mathbb{E}\left[\|\nabla_\theta V(\theta_t)\|^2\right] + \frac{1 + \lambda}{2} \left\|\mathbb{E}\left[u_t(\omega_t) - (F(\theta_t) + \lambda I)^{-1} \nabla_\theta V(\theta_t)\right]\right\|^2\right)$$

$$- L_V \beta_t^2 \mathbb{E}\left[\left\|u_t(\omega_t) - (F(\theta_t) + \lambda I)^{-1} \nabla_\theta V(\theta_t)\right\|^2\right] - \frac{L_V \beta_t^2}{\lambda^2} \mathbb{E}\left[\|\nabla_\theta V(\theta_t)\|^2\right]$$

$$=\mathbb{E}\left[V(\theta_t)\right] + \left(\frac{\beta_t}{2(1+\lambda)} - \frac{L_V\beta_t^2}{\lambda^2}\right)\mathbb{E}\left[\|\nabla_\theta V(\theta_t)\|^2\right]$$
$$- \left(\frac{\beta_t(1+\lambda)}{2}\right)\left\|\mathbb{E}\left[u_t(\omega_t) - (F(\theta_t)+\lambda I)^{-1}\nabla_\theta V(\theta_t)\right]\right\|^2$$
$$- L_V\beta_t^2\mathbb{E}\left[\left\|u_t(\omega_t) - (F(\theta_t)+\lambda I)^{-1}\nabla_\theta V(\theta_t)\right\|^2\right].$$

Denote $v_t(\omega_t) = \phi(s_t, a_t)^T\omega_t\frac{d}{d\theta}\log\pi_{\theta_t}(a_t \mid s_t)$.

For the term $\left\|u_t(\omega_t) - (F(\theta_t)+\lambda I)^{-1}\nabla_\theta V(\theta_t)\right\|^2$, we have that

$$\left\|u_t(\omega_t) - (F(\theta_t)+\lambda I)^{-1}\nabla_\theta V(\theta_t)\right\|^2$$
$$= \left\|u_t(\omega_t) - (F(\theta_t)+\lambda I)^{-1}v_t(\omega_t) + (F(\theta_t)+\lambda I)^{-1}v_t(\omega_t) - (F(\theta_t)+\lambda I)^{-1}\nabla_\theta V(\theta_t)\right\|^2$$
$$\leq 2\left\|u_t(\omega_t) - (F(\theta_t)+\lambda I)^{-1}v_t(\omega_t)\right\|^2 + 2\left\|(F(\theta_t)+\lambda I)^{-1}v_t(\omega_t) - (F(\theta_t)+\lambda I)^{-1}\nabla_\theta V(\theta_t)\right\|^2$$
$$= 2\left\|\left[(F_t+\lambda I)^{-1} - (F(\theta_t)+\lambda I)^{-1}\right]v_t(\omega_t)\right\|^2 + 2\left\|(F(\theta_t)+\lambda I)^{-1}(v_t(\omega_t) - \nabla_\theta V(\theta_t))\right\|^2$$
$$= 2\left\|\left[(F_t+\lambda I)^{-1} - (F(\theta_t)+\lambda I)^{-1}\right](v_t(\omega_t) - \nabla_\theta V(\theta_t) + \nabla_\theta V(\theta_t))\right\|^2$$
$$+ 2\left\|(F(\theta_t)+\lambda I)^{-1}(v_t(\omega_t) - \nabla_\theta V(\theta_t))\right\|^2$$
$$\leq 4\left\|\left[(F_t+\lambda I)^{-1} - (F(\theta_t)+\lambda I)^{-1}\right](v_t(\omega_t) - \nabla_\theta V(\theta_t))\right\|^2$$
$$+ 4\left\|\left[(F_t+\lambda I)^{-1} - (F(\theta_t)+\lambda I)^{-1}\right]\nabla_\theta V(\theta_t)\right\|^2$$
$$+ 2\left\|(F(\theta_t)+\lambda I)^{-1}(v_t(\omega_t) - \nabla_\theta V(\theta_t))\right\|^2$$
$$\leq \left[4\left\|(F_t+\lambda I)^{-1} - (F(\theta_t)+\lambda I)^{-1}\right\|^2 + 2\left\|(F(\theta_t)+\lambda I)^{-1}\right\|^2\right]\|v_t(\omega_t) - \nabla_\theta V(\theta_t)\|^2$$
$$+ 4\left\|(F_t+\lambda I)^{-1} - (F(\theta_t)+\lambda I)^{-1}\right\|^2\|\nabla_\theta V(\theta_t)\|^2$$
$$\leq \left[8\left\|(F_t+\lambda I)^{-1}\right\|^2 + 10\left\|(F(\theta_t)+\lambda I)^{-1}\right\|^2\right]\|v_t(\omega_t) - \nabla_\theta V(\theta_t)\|^2$$
$$+ 4\left\|(F_t+\lambda I)^{-1} - (F(\theta_t)+\lambda I)^{-1}\right\|^2\|\nabla_\theta V(\theta_t)\|^2$$
$$\leq \frac{18}{\lambda^2}\|v_t(\omega_t) - \nabla_\theta V(\theta_t)\|^2 + 4\left\|(F_t+\lambda I)^{-1} - (F(\theta_t)+\lambda I)^{-1}\right\|^2\|\nabla_\theta V(\theta_t)\|^2$$
$$= \frac{18}{\lambda^2}\|v_t(\omega_t) - \nabla_\theta V(\theta_t)\|^2 + 4\left\|(F_t+\lambda I)^{-1}(F_t - F(\theta_t))(F(\theta_t)+\lambda I)^{-1}\right\|^2\|\nabla_\theta V(\theta_t)\|^2$$
$$\leq \frac{18}{\lambda^2}\|v_t(\omega_t) - \nabla_\theta V(\theta_t)\|^2 + 4\left\|(F_t+\lambda I)^{-1}\right\|^2\|(F_t - F(\theta_t))\|^2\left\|(F(\theta_t)+\lambda I)^{-1}\right\|^2\|\nabla_\theta V(\theta_t)\|^2$$
$$\leq \frac{18}{\lambda^2}\|v_t(\omega_t) - \nabla_\theta V(\theta_t)\|^2 + \frac{4}{\lambda^2(1-\gamma)^2}\|F_t - F(\theta_t)\|^2.$$

For the term $\|v_t(\omega_t) - \nabla_\theta V(\theta_t)\|^2$, we have that

$$\|v_t(\omega_t) - \nabla_\theta V(\theta_t)\|^2 \leq 3L_g^2\|\Delta_t\|^2 + 3\delta^2 + 3\sigma_a^2.$$

Rearranging the term, we then have

$$\left(\frac{\beta_t}{2(1+\lambda)} - \frac{L_V\beta_t^2}{\lambda^2}\right)\|\nabla_\theta V(\theta_t)\|^2$$
$$\leq V(\theta_t) - V(\theta_{t+1})$$
$$+ \left(\frac{\beta_t(1+\lambda)}{2}\right)\left(\frac{18}{\lambda^2}\left(2L_g^2\|\Delta_t\|^2 + 2\delta^2\right) + \frac{4}{\lambda^2(1-\gamma)^2}\|F_t - F(\theta_t)\|^2\right)$$
$$+ L_V\beta_t^2\left(\frac{18}{\lambda^2}\left(3L_g^2\|\Delta_t\|^2 + 3\delta^2 + 3\sigma_a^2\right) + \frac{4}{\lambda^2(1-\gamma)^2}\|F_t - F(\theta_t)\|^2\right).$$

Using the assumption that $\frac{L_V \beta_t^2}{\lambda^2} < \frac{\beta_t}{4(\lambda+1)}$ and summing over $t = T/2, \cdots, T-1$, we have that

$$
\sum_{t=T/2}^{T-1} \frac{\beta_t}{4(\lambda+1)} \mathbb{E}\left[\|\nabla_t\|^2\right] \leq \mathbb{E}\left[V(\theta_{T/2})\right] - \mathbb{E}\left[V(\theta_T)\right] + \frac{36 L_g^2}{\lambda^2} \sum_{t=T/2}^{T-1} \beta_t \mathbb{E}\left[\|\Delta_t\|^2\right]
$$

$$
+ \frac{36\delta^2}{\lambda^2} \sum_{t=T/2}^{T-1} \beta_t + \frac{18 L_V \sigma_a^2}{\lambda^2} \sum_{t=T/2}^{T-1} \beta_t
$$

$$
+ \frac{8}{\lambda^2(1-\gamma)^2} \sum_{t=T/2}^{T-1} \beta_t \mathbb{E}\left[\|F_t - F(\theta_t)\|^2\right].
$$

Dividing by $(\beta_T/4(\lambda+1))(T/2)$ we get

$$
\frac{2}{T} \sum_{t=T/2}^{T-1} \mathbb{E}\left[\|\nabla_t\|^2\right] \leq \frac{16 \mathbb{E}\left[V(\theta_{T/2})\right] - 16\mathbb{E}\left[V(\theta_T)\right]}{\beta_T T} + \frac{36 c_\beta L_g^2}{\lambda^2 T/2} \sum_{t=T/2}^{T-1} \mathbb{E}\left[\|\Delta_t\|^2\right] + \frac{36 c_\beta \delta^2}{\lambda^2}
$$

$$
+ \frac{18 c_\beta \beta_T L_V \sigma_a^2}{\lambda^2} + \frac{8 c_\beta}{\lambda^2(1-\gamma)^2 T/2} \sum_{t=T/2}^{T-1} \mathbb{E}\left[\|F_t - F(\theta_t)\|^2\right].
$$

## A.4 ANALYSIS FOR FISHER INFORMATION UPDATE

**Proof of Lemma 5** Recall the update of the estimated fisher information matrix: $F_{t+1} = (1 - \zeta_t)F_t + \zeta_t \nabla \log \pi(a_t|s_t) \nabla \log \pi(a_t|s_t)^T$, then we have

$$
\mathbb{E}\left[\|F_{t+1} - F(\theta_{t+1})\|^2\right]
$$

$$
= \mathbb{E}\left[\|(1-\zeta_t)F_t + \zeta_t \nabla \log \pi(a_t|s_t) \nabla \log \pi(a_t|s_t)^T - F(\theta_{t+1})\|^2\right]
$$

$$
= \mathbb{E}\left[\|(1-\zeta_t)(F_t - F(\theta_t)) + \zeta_t\left(\nabla \log \pi(a_t|s_t) \nabla \log \pi(a_t|s_t)^T - F(\theta_{t+1})\right) - (1-\zeta_t)(F(\theta_{t+1}) - F(\theta_t))\|^2\right]
$$

$$
\leq (1-\zeta_t)^2 \mathbb{E}\left[\|F_t - F(\theta_t)\|^2\right] + \zeta_t^2 \mathbb{E}\left[\|\nabla \log \pi(a_t|s_t) \nabla \log \pi(a_t|s_t)^T - F(\theta_{t+1})\|^2\right] + (1-\zeta_t)^2 \mathbb{E}\left[\|F(\theta_{t+1}) - F(\theta_t)\|^2\right]
$$

$$
+ 2(1-\zeta_t)^2 \mathbb{E}\left[\langle F_t - F(\theta_t), F(\theta_{t+1}) - F(\theta_t)\rangle\right] + 2\zeta_t(1-\zeta_t)\mathbb{E}\left[\langle F_t - F(\theta_t), \nabla \log \pi(a_t|s_t) \nabla \log \pi(a_t|s_t)^T - F(\theta_{t+1})\rangle\right]
$$

$$
+ 2\zeta_t(1-\zeta_t)\mathbb{E}\left[\langle F(\theta_{t+1}) - F(\theta_t), \nabla \log \pi(a_t|s_t) \nabla \log \pi(a_t|s_t)^T - F(\theta_{t+1})\rangle\right]
$$

$$
\leq (1-\zeta_t)\mathbb{E}\left[\|F_t - F(\theta_t)\|^2\right] + \zeta_t^2 \sigma_F^2 + L_F \mathbb{E}\left[\|\theta_{t+1} - \theta_t\|^2\right] + 2(1-\zeta_t)^2 \mathbb{E}\left[\langle F_t - F(\theta_t), F(\theta_{t+1}) - F(\theta_t)\rangle\right].
$$

For term $\langle F_t - F(\theta_t), F(\theta_{t+1}) - F(\theta_t)\rangle$, we adopt the same analysis as the critic update and can derive the upper bound:

$$
\langle F_t - F(\theta_t), F(\theta_{t+1}) - F(\theta_t)\rangle
$$

$$
\leq L_F \beta_t \lambda^{-1} \|g(\theta_t, \omega_t)\| \|F_t - F(\theta_t)\| + \beta_t^2 \lambda^{-2} \lambda' \sigma_{a'}^2 \|F_t - F(\theta_t)\|
$$

$$
\leq L_F \beta_t \lambda^{-1} \|g(\theta_t, \omega_t)\| \|F_t - F(\theta_t)\| + \beta_t^2 \lambda^{-2} \lambda' \sigma_{a'}^2 \sigma_F.
$$

Similar to the critic update, we use the recursion form to derive the final bound of the estimated fisher information matrix error.

$$
\frac{2}{T} \sum_{t=T/2}^{T-1} \mathbb{E}\left[\|F_t - F(\theta_t)\|^2\right] \leq \frac{1}{\zeta_T T}\left\|F_{T/2} - F(\theta_{T/2})\right\|^2 + c_\zeta^2 \zeta_T \sigma_F^2
$$

$$
+ 2L_F^2 \frac{c_\beta^2 \beta_T^2 \lambda^{-2}}{\zeta_T} \sigma_a^2 + 2L_F^2 \frac{c_\beta^2 \beta_T^2 \lambda^{-2}}{\zeta_T} \bar{\delta}^2 + \frac{c_\beta^2 \beta_T^2 \lambda^2 \lambda' \sigma_{a'}^2 \sigma_F}{\zeta_T}
$$

$$
+ \frac{c_\beta \beta_T \lambda^{-1} L_F}{\zeta_T} \frac{2}{T} \sum_{t=T/2}^{T-1} \mathbb{E}\left[\|\nabla_t\|^2\right]
$$

$$+ \frac{c_\beta \beta_T \lambda^{-1} L_g L_F}{\zeta_T} \frac{2}{T} \sum_{t=T/2}^{T-1} \mathbb{E}\left[\|\|\Delta_t\|\|\right]^2 \ .$$

$$\|F_{t+1} - F(\theta_{t+1})\|^2$$
$$= \left\|(1 - \zeta_t)F_t + \zeta_t \nabla \log \pi(a_t|s_t)\nabla \log \pi(a_t|s_t)^T - F(\theta_{t+1})\right\|^2$$
$$= \left\|(1 - \zeta_t)(F_t - F(\theta_t)) + \zeta_t \left(\nabla \log \pi(a_t|s_t)\nabla \log \pi(a_t|s_t)^T - F(\theta_{t+1})\right) - (1 - \zeta_t)(F(\theta_{t+1}) - F(\theta_t))\right\|^2$$
$$\leq (1 - \zeta_t)\|F_t - F(\theta_t)\|^2 + \zeta_t^2 \left\|\nabla \log \pi(a_t|s_t)\nabla \log \pi(a_t|s_t)^T - F(\theta_{t+1})\right\|^2 + (1 - \zeta_t)\|F(\theta_{t+1}) - F(\theta_t)\|^2$$
$$\leq (1 - \zeta_t)\|F_t - F(\theta_t)\|^2 + \zeta_t^2 \sigma_F^2 + L_F^2 \|\theta_{t+1} - \theta_t\|^2$$
$$\leq (1 - \zeta_t)\|F_t - F(\theta_t)\|^2 + \zeta_t^2 \sigma_F^2$$
$$\quad + 2L_F^2 \beta_t^2 \lambda^{-2}(\sigma_a^2 + 3L_g^2 \|\Delta_t\|^2 + 3\|\nabla_t\|^2 + 3\bar{\delta}^2) \ .$$

Similar to the critic update, we can use this recursion form to derive the final bound of the estimated fisher information matrix error

$$\frac{2}{T} \sum_{t=T/2}^{T-1} \mathbb{E}\left[\|F_t - F(\theta_t)\|^2\right] \leq \frac{1}{\zeta_T T}\left\|F_{T/2} - F(\theta_{T/2})\right\|^2 + c_\zeta^2 \zeta_T \sigma_F^2$$
$$+ 2L_F^2 \frac{c_\beta^2 \beta_T^2 \lambda^{-2}}{\zeta_T}\sigma_a^2 + 2L_F^2 \frac{c_\beta^2 \beta_T^2 \lambda^{-2}}{\zeta_T}\bar{\delta}^2$$
$$+ \frac{c_\beta \beta_T \lambda^{-1} L_F}{\zeta_T} \frac{2}{T} \sum_{t=T/2}^{T-1} \mathbb{E}\left[\|\|\nabla_t\|\|\right]^2$$
$$+ \frac{c_\beta \beta_T \lambda^{-1} L_g L_F}{\zeta_T} \frac{2}{T} \sum_{t=T/2}^{T-1} \mathbb{E}\left[\|\|\Delta_t\|\|\right]^2 \ .$$

## A.5 Attain stationary point convergence

*Proof of Theorem 1.* We perform an interconnected iteration system analysis. We choose $\alpha_t = O(1/\sqrt{t})$, $\beta_t = O(1/\sqrt{t})$, $\zeta_t = O(1/\sqrt{t})$, and denote $c' = \beta_t/\alpha_t$, $c'' = \beta_t/\gamma_t$. We define $X_T$, $Y_T$ and $Z_T$, which denote the expectation of the critic error, the square norm of the policy gradient, and the Fisher information estimated error, respectively:

$$X_T = \frac{2}{T} \sum_{t=T/2}^{T-1} \mathbb{E}\left[\|\Delta_t\|^2\right], Y_T = \frac{2}{T} \sum_{t=T/2}^{T-1} \mathbb{E}\left[\|\nabla_t\|^2\right], Z_T = \frac{2}{T} \sum_{t=T/2}^{T-1} \mathbb{E}\left[\|e_t\|^2\right] \ .$$

Then we have that

$$X_T \leq O\left(\frac{1}{\sqrt{T}}\right) + C_1 \sqrt{X_T} + C_2 Y_T \ ,$$

$$Y_T \leq O\left(\frac{1}{\sqrt{T}}\right) + C_3 X_T + C_4 Z_T + C_5 \ ,$$

$$Z_T \leq O\left(\frac{1}{\sqrt{T}}\right) + C_6 X_T + C_7 Y_T \ ,$$

where $C_1 = \frac{8c_\beta c' \lambda^{-1} L_\omega \delta}{\mu}$, $C_2 = \frac{8c_\beta c' \lambda^{-1} L_\omega}{\mu}$, $C_3 = \frac{36c_\beta L_g^2}{\lambda^2}$, $C_4 = \frac{8c_\beta}{\lambda^2(1-\gamma)^2}$, $C_5 = \frac{36c_\beta \delta^2}{\lambda^2}$, $C_6 = c''c_\beta \lambda^{-1} L_g L_F$, $C_7 = c''c_\beta \lambda^{-1} L_F$.

For $Y_T$, we can further imply that

$$Y_T \leq O\left(\frac{1}{\sqrt{T}}\right) + C_3 X_T + C_4 C_6 X_T + C_4 C_7 Y_T + C_5 \ .$$

Rearranging the term we have if $1 - C_4 C_7 > 0$, then

$$Y_T \leq O\left(\frac{1}{\sqrt{T}}\right) + \frac{C_3 + C_4 C_6}{1 - C_4 C_7} X_T + \frac{C_5}{1 - C_4 C_7}.$$

Denote $C_3' = \frac{C_3 + C_4 C_6}{1 - C_4 C_7}$, $C_5' = \frac{C_5}{1 - C_4 C_7}$. Let us substitute $X_T = x^2$, then

$$x^2 \leq O\left(\frac{1}{\sqrt{T}}\right) + C_1 x + C_2 Y_T.$$

We can upper bound $x$ by the largest root of this quadratic:

$$x \leq \frac{C_1 + \sqrt{C_1^2 + 4(O(1/\sqrt{T}) + C_2 Y_T)}}{2}.$$

Squaring both sides and using the inequality $(x + y)^2 \leq 2x^2 + 2y^2$, we have

$$x^2 \leq \frac{2C_1^2 + 2(C_1^2 + 4(O(1/\sqrt{T}) + C_2 Y_T))}{4},$$

and then using $X_t = x^2$ we obtain

$$X_T \leq O\left(\frac{1}{\sqrt{T}}\right) + 2C_2 Y_T + C_1^2.$$

Now use the bound of $Y_T$ we have

$$X_T \leq O\left(\frac{1}{\sqrt{T}}\right) + 2C_2 C_3' X_T + C_2 C_5'.$$

Rearranging the term, if $1 - 2C_2 C_3' > 0$, we have

$$X_T \leq O\left(\frac{1}{\sqrt{T}}\right) + \frac{C_2 C_5'}{1 - 2C_2 C_3'},$$

where $\frac{C_2 C_5'}{1 - 2C_2 C_3'} = O(\delta^2)$. We can then get the final result by using this bound to control $Y_T$ and $Z_T$.

At last, we need to select appropriate step sizes such that $1 - C_4 C_7 > 0$ and $1 - 2C_2 C_3' > 0$ hold. These two conditions can be satisfied by selecting $c', c''$ small enough. More specifically, to guarantee $1 - C_4 C_7 > 1/2$, we can let $c'' < \frac{\lambda^3 (1-\gamma)^2}{16 c_\beta^2 L_g L_F}$. To guarantee $1 - 2C_2 C_3' > 0$, we can let $c' \leq \min\{\frac{\lambda^4 (1-\gamma)^2 \mu}{128 c_\beta^3 L_g L_F L_\omega}, \frac{\lambda^3 \mu}{288 c_\beta^2 L_g^2 L_\omega}\}$. This can be ensured by making the step size of the actor update small enough. $\qquad \square$

### A.6 ATTAIN GLOBAL OPTIMALITY CONVERGENCE

**Proof of Theorem 2** Given the above convergence result on the gradient norm, we proceed to prove the convergence of NAC in terms of the function value. Denote $D(\theta) = D_{KL}(\pi^*(\cdot|s), \pi(\cdot|s))$, and denote $L_\psi = K_1$. Then we proceed as follows:

$$D(\theta_t) - D(\theta_{t+1})$$

$$= \mathbb{E}_* \left[ \log(\pi_{\theta_{t+1}}(a|s)) - \log \pi_{\theta_t}(a|s) \right]$$

$$\geq \mathbb{E}_* \left[ \nabla_\theta \log \pi_{\theta_t}(a|s) \right]^T (\theta_{t+1} - \theta_t) - \frac{L_\psi}{2} \| \theta_{t+1} - \theta_t \|^2$$

$$= \beta_t \mathbb{E}_* \left[ \nabla_\theta \log \pi_{\theta_t}(a|s) \right]^T u_t(\omega_t) - \frac{L_\psi \beta_t^2}{2} \| u_t(\omega_t) \|^2$$

$$= \beta_t \mathbb{E}_* \left[ \nabla_\theta \log \pi_{\theta_t}(a|s) \right]^T u_{\theta_t, \lambda} + \beta_t \mathbb{E}_* \left[ \nabla_\theta \log \pi_{\theta_t}(a|s) \right]^T (u_t(\omega_t) - u_{\theta_t, \lambda}) - \frac{L_\psi \beta_t^2}{2} \| u_t(\omega_t) \|^2$$

$$= \beta_t \mathbb{E}_* \left[ \nabla_\theta \log \pi_{\theta_t}(a|s) \right]^T u_{\theta_t}^\dagger + \beta_t \mathbb{E}_* \left[ \nabla_\theta \log \pi_{\theta_t}(a|s) \right]^T (u_{\theta_t, \lambda} - u_{\theta_t}^\dagger)$$

$$+ \beta_t \mathbb{E}_* \left[ \nabla_\theta \log \pi_{\theta_t}(a|s) \right]^T (u_t(\omega_t) - u_{\theta_t, \lambda}) - \frac{L_\psi \beta_t^2}{2} \| u_t(\omega_t) \|^2$$

$$
\begin{aligned}
=&\beta_t \mathbb{E}_* \left[ A_{\pi_{\theta_t}(s,a)} \right] + \beta_t \mathbb{E}_* \left[ \nabla_\theta \log \pi_{\theta_t}(a|s) \right]^T (u_{\theta_t,\lambda} - u_{\theta_t}^\dagger) \\
&+ \beta_t \mathbb{E}_* \left[ \nabla_\theta \log \pi_{\theta_t}(a|s) \right]^T (u_t(\omega_t) - u_{\theta_t,\lambda}) - \frac{L_\psi \beta_t^2}{2} \| u_t(\omega_t) \|^2 \\
&+ \beta_t \mathbb{E}_* \left[ \nabla_\theta \log \pi_{\theta_t}(a|s)^T u_{\theta_t}^\dagger - A_{\pi_{\theta_t}(s,a)} \right] \\
=&(1-\gamma)\beta_t \left( V^* - V^{\pi_{\theta_t}} \right) + \beta_t \mathbb{E}_* \left[ \nabla_\theta \log \pi_{\theta_t}(a|s) \right]^T (u_{\theta_t,\lambda} - u_{\theta_t}^\dagger) \\
&+ \beta_t \mathbb{E}_* \left[ \nabla_\theta \log \pi_{\theta_t}(a|s) \right]^T (u_t(\omega_t) - u_{\theta_t,\lambda}) - \frac{L_\psi \beta_t^2}{2} \| u_t(\omega_t) \|^2 \\
&+ \beta_t \mathbb{E}_* \left[ \nabla_\theta \log \pi_{\theta_t}(a|s)^T u_{\theta_t}^\dagger - A_{\pi_{\theta_t}(s,a)} \right] \\
\geq&(1-\gamma)\beta_t \left( V^* - V^{\pi_{\theta_t}} \right) + \beta_t \mathbb{E}_* \left[ \nabla_\theta \log \pi_{\theta_t}(a|s) \right]^T (u_{\theta_t,\lambda} - u_{\theta_t}^\dagger) \\
&+ \beta_t \mathbb{E}_* \left[ \nabla_\theta \log \pi_{\theta_t}(a|s) \right]^T (u_t(\omega_t) - u_{\theta_t,\lambda}) - \frac{L_\psi \beta_t^2}{2} \| u_t(\omega_t) \|^2 \\
&- \beta_t \sqrt{ \mathbb{E}_* \left[ \nabla_\theta \log \pi_{\theta_t}(a|s)^T u_{\theta_t}^\dagger - A_{\pi_{\theta_t}(s,a)} \right]^2 } \\
\geq&(1-\gamma)\beta_t \left( V^* - V^{\pi_{\theta_t}} \right) + \beta_t \mathbb{E}_* \left[ \nabla_\theta \log \pi_{\theta_t}(a|s) \right]^T (u_{\theta_t,\lambda} - u_{\theta_t}^\dagger) \\
&+ \beta_t \mathbb{E}_* \left[ \nabla_\theta \log \pi_{\theta_t}(a|s) \right]^T (u_t(\omega_t) - u_{\theta_t,\lambda}) - \frac{L_\psi \beta_t^2}{2} \| u_t(\omega_t) \|^2 \\
&- \sqrt{ \left\| \frac{\mu_{\pi^*}}{\mu_{\pi_{\theta_t}}} \right\|_\infty } \beta_t \sqrt{ \mathbb{E}_{\pi_{\theta_t}} \left[ \nabla_\theta \log \pi_{\theta_t}(a|s)^T u_{\theta_t}^\dagger - A_{\pi_{\theta_t}(s,a)} \right]^2 } \\
\geq&(1-\gamma)\beta_t \left( V^* - V^{\pi_{\theta_t}} \right) + \beta_t \mathbb{E}_* \left[ \nabla_\theta \log \pi_{\theta_t}(a|s) \right]^T (u_{\theta_t,\lambda} - u_{\theta_t}^\dagger) \\
&+ \beta_t \mathbb{E}_* \left[ \nabla_\theta \log \pi_{\theta_t}(a|s) \right]^T (u_t(\omega_t) - u_{\theta_t,\lambda}) - \frac{L_\psi \beta_t^2}{2} \| u_t(\omega_t) \|^2 \\
&- \sqrt{ \frac{1}{1-\gamma} \left\| \frac{\mu_{\pi^*}}{\mu_{\pi_{\theta_0}}} \right\|_\infty } \beta_t \sqrt{ \mathbb{E}_{\pi_{\theta_t}} \left[ \nabla_\theta \log \pi_{\theta_t}(a|s)^T u_{\theta_t}^\dagger - A_{\pi_{\theta_t}(s,a)} \right]^2 } \\
\geq&(1-\gamma)\beta_t \left( V^* - V^{\pi_{\theta_t}} \right) - \beta_t C_r \lambda - \beta_t \mathbb{E}_* \left[ \nabla_\theta \log \pi_{\theta_t}(a|s) \right]^T (u_t(\omega_t) - u_{\theta_t,\lambda}) \\
&- \sqrt{ \frac{1}{1-\gamma} \left\| \frac{\mu_{\pi^*}}{\mu_{\pi_{\theta_0}}} \right\|_\infty } \beta_t \sqrt{ \mathbb{E}_{\pi_{\theta_t}} \left[ \nabla_\theta \log \pi_{\theta_t}(a|s)^T u_{\theta_t}^\dagger - A_{\pi_{\theta_t}(s,a)} \right]^2 } - \frac{L_\psi \beta_t^2}{2} \| u_t(\omega_t) \|^2 \\
\geq&(1-\gamma)\beta_t \left( V^* - V^{\pi_{\theta_t}} \right) - \beta_t C_r \lambda - \beta_t \mathbb{E}_* \left[ \nabla_\theta \log \pi_{\theta_t}(a|s) \right]^T (u_t(\omega_t) - u_{\theta_t,\lambda}) \\
&- \sqrt{ \frac{1}{1-\gamma} \left\| \frac{\mu_{\pi^*}}{\mu_{\pi_{\theta_0}}} \right\|_\infty } \beta_t \delta' - \frac{L_\psi \beta_t^2}{2} \| u_t(\omega_t) \|^2 \\
\geq&(1-\gamma)\beta_t \left( V^* - V^{\pi_{\theta_t}} \right) - \beta_t C_r \lambda - \beta_t \mathbb{E}_* \left[ \nabla_\theta \log \pi_{\theta_t}(a|s) \right]^T (u_t(\omega_t) - u_{\theta_t,\lambda}) \\
&- \sqrt{ \frac{1}{1-\gamma} \left\| \frac{\mu_{\pi^*}}{\mu_{\pi_{\theta_0}}} \right\|_\infty } \beta_t \delta' - L_\psi \beta_t^2 \| u_t(\omega_t) - u_{\theta_t,\lambda} \|^2 - L_\psi \beta_t^2 \| u_{\theta_t,\lambda} \|^2 \\
\geq&(1-\gamma)\beta_t \left( V^* - V^{\pi_{\theta_t}} \right) - \beta_t C_r \lambda - \beta_t \mathbb{E}_* \left[ \nabla_\theta \log \pi_{\theta_t}(a|s) \right]^T (u_t(\omega_t) - u_{\theta_t,\lambda}) \\
&- \sqrt{ \frac{1}{1-\gamma} \left\| \frac{\mu_{\pi^*}}{\mu_{\pi_{\theta_0}}} \right\|_\infty } \beta_t \delta' - L_\psi \beta_t^2 \| u_t(\omega_t) - u_{\theta_t,\lambda} \|^2 - \frac{L_\psi \beta_t^2}{\lambda^2} \| u_{\theta_t,\lambda} \|^2 \ .
\end{aligned}
$$

Rearranging the equation, we have

$$
V^* - \mathbb{E}\left[ V(\theta_t) \right]
$$

$$\leq \frac{\mathbb{E}\left[D(\theta_t)\right] - \mathbb{E}\left[D(\theta_{t+1})\right]}{(1-\gamma)\beta_t} + \sqrt{\frac{1}{(1-\gamma)^3}\left\|\frac{\mu_{\pi^*}}{\mu_{\pi_{\theta_0}}}\right\|_\infty}\delta'$$

$$+ \frac{L_\psi\beta_t\mathbb{E}\left[\|u_t(\omega_t) - u_{\theta_t,\lambda}\|^2\right]}{1-\gamma} + \frac{L_\psi\beta_t}{(1-\gamma)\lambda^2}\mathbb{E}\left[\|\nabla_\theta V(\theta_t)\|^2\right] + \frac{C_r\lambda}{1-\gamma}$$

$$+ \frac{\mathbb{E}\left[\mathbb{E}_*\left[\nabla_\theta \log \pi_{\theta_t}(a|s)\right]^T (u_t(\omega_t) - u_{\theta_t,\lambda})\right]}{1-\gamma}$$

$$\leq \frac{\mathbb{E}\left[D(\theta_t)\right] - \mathbb{E}\left[D(\theta_{t+1})\right]}{(1-\gamma)\beta_t} + \sqrt{\frac{1}{(1-\gamma)^3}\left\|\frac{\mu_{\pi^*}}{\mu_{\pi_{\theta_0}}}\right\|_\infty}\delta'$$

$$+ \frac{L_\psi\beta_t\mathbb{E}\left[\|u_t(\omega_t) - u_{\theta_t,\lambda}\|^2\right]}{1-\gamma} + \frac{L_\psi\beta_t}{(1-\gamma)\lambda^2}\mathbb{E}\left[\|\nabla_\theta V(\theta_t)\|^2\right] + \frac{C_r\lambda}{1-\gamma}$$

$$+ \frac{\|\mathbb{E}\left[u_t(\omega_t) - u_{\theta_t,\lambda}\right]\|}{1-\gamma}.$$

Recall that for $\mathbb{E}\left[\|u_t(\omega_t) - u_{\theta_t,\lambda}\|^2\right]$, we have the bound

$$\mathbb{E}\left[\|u_t(\omega_t) - u_{\theta_t,\lambda}\|^2\right] \leq \frac{18}{\lambda^2}\left(3L_g^2\mathbb{E}\left[\|\Delta_t\|^2\right] + 3\delta^2 + 3\sigma_a^2\right) + \frac{4}{\lambda^2(1-\gamma)^2}\mathbb{E}\left[\|F_t - F(\theta_t)\|^2\right].$$

We also have the bound

$$\|\mathbb{E}\left[u_t(\omega_t) - u_{\theta_t,\lambda}\right]\|^2 \leq \frac{18}{\lambda^2}\left(3L_g^2\mathbb{E}\left[\|\Delta_t\|^2\right] + 3\delta^2\right) + \frac{4}{\lambda^2(1-\gamma)^2}\mathbb{E}\left[\|F_t - F(\theta_t)\|^2\right].$$

Summing over $T/2, T/2 + 1, \cdots, T - 1$ yields

$$V^* - \frac{2}{T}\sum_{t=T/2}^{T-1}\mathbb{E}\left[V(\theta_t)\right]$$

$$\leq \frac{D_{\max}}{(1-\gamma)\beta_T} + \sqrt{\frac{1}{(1-\gamma)^3}\left\|\frac{\mu_{\pi^*}}{\mu_{\pi_{\theta_0}}}\right\|_\infty}\delta' + \frac{L_\psi\beta_{T/2}\frac{2}{T}\sum_{t=T/2}^{T-1}\mathbb{E}\left[\|u_t(\omega_t) - u_{\theta_t,\lambda}\|^2\right]}{1-\gamma}$$

$$+ \frac{L_\psi\beta_{T/2}\frac{2}{T}\sum_{t=T/2}^{T-1}\mathbb{E}\left[\|\nabla_t\|^2\right]}{(1-\gamma)\lambda^2} + \frac{C_r\lambda}{1-\gamma} + \frac{2\sqrt{T}\sqrt{\sum_{t=T/2}^{T-1}\|\mathbb{E}\left[u_t(\omega_t) - u_{\theta_t,\lambda}\right]\|^2}}{(1-\gamma)T}$$

$$\leq O\left(\delta^2 + \delta' + T^{-1/4} + \lambda\right).$$

The first inequality is due to the Cauchy-Schwarz inequality. In the last inequality, we use the results in Theorem 1 to obtain the final bound. If we set $\lambda = O(\delta^2 + \delta')$, then the result is what we desire, and thus the proof is completed.

## B ANALYSIS FOR MARKOVIAN SAMPLE

Indeed the i.i.d assumption in the update is too strong and needs a simulator to output a stationary $(s, a)$ sample at each step, a more natural approach is to use the Markovian sampling at each step. The major difference of Markovian from i.i.d. sampling is that at the $t$-th iteration, the state $s_t$ is evolving the Markov chain instead of sampling from the stationary distribution. More specifically, at time $t$, the state $s_t$ is induced by the distribution $P(\cdot \mid s_{t-1}, a_{t-1})$, where $(s_{t-1}, a_{t-1})$ are samples used in the $t - 1$ step. Then we follow policy $\pi_{\theta_t}$ and take an action $a_t$. Then we use $(s_t, a_t)$ to update the actor, critic, and estimated Fisher information matrix. Then the next state $s_{t+1}$ is induced from $P(\cdot \mid s_t, a_t)$ and will be used in the next update.

The main challenge in applying the Markovian sample is that the estimated gradient $\nabla_\theta \log \pi_{\theta_t}(s_t, a_t)$, the feature $\phi(s_t, a_t)$ which use $(s_t, a_t)$ sample are no longer the unbiased estimation of the true gradient and feature. Thus we need to control the distance between the distribution of the Markovian sample and the distribution of the stationary sample. More specifically, we need to bound the total variant distance between those two distributions.

Similar to Chen & Zhao (2022), We make use of the following auxiliary Markov chain to deal with the Markovian noise.

Auxiliary Markov Chain:

$$s_{t-\tau} \overset{\theta_{t-\tau}}{\to} a_{t-\tau} \overset{P}{\to} s_{t-\tau+1} \overset{\theta_{t-\tau}}{\to} \tilde{a}_{t-\tau+1} \overset{P}{\to} \tilde{s}_{t-\tau+2} \overset{\theta_{t-\tau}}{\to} \tilde{a}_{t-\tau+2} \cdots \overset{P}{\to} \tilde{s}_t \overset{\theta_{t-\tau}}{\to} \tilde{a}_t$$

Original Markov Chain:

$$s_{t-\tau} \overset{\theta_{t-\tau}}{\to} a_{t-\tau} \overset{P}{\to} s_{t-\tau+1} \overset{\theta_{t-\tau+1}}{\to} a_{t-\tau+1} \overset{P}{\to} s_{t-\tau+2} \overset{\theta_{t-\tau+2}}{\to} a_{t-\tau+2} \cdots \overset{P}{\to} s_t \overset{\theta_t}{\to} a_t$$

We denote that the auxiliary Markov chain $(\tilde{s}_t, \tilde{a}_t)$ is induced from the distribution $\tilde{\mu}_t$, the original Markov chain $(s_t, a_t)$ is induced from the distribution $\mu_t$, and recall that the stationary distribution is $\mu_{\theta t}$.

We then show the total variant distance between these distributions, conditioned on $s_{t-\tau+1}, \theta_{t-\tau}$.

$$d_{TV}(\mu_t, \tilde{\mu}_t)$$
$$= d_{TV}(\mathbb{P}(s_t, a_t \in \cdot | s_{t-\tau+1}, \theta_{t-\tau}), \mathbb{P}(\tilde{s}_t, \tilde{a}_t \in \cdot | s_{t-\tau+1}, \theta_{t-\tau}))$$
$$\leq d_{TV}(\mathbb{P}(s_t \in \cdot | s_{t-\tau+1}, \theta_{t-\tau}), \mathbb{P}(\tilde{s}_t \in \cdot | s_{t-\tau+1}, \theta_{t-\tau})) + \frac{1}{2} L_\pi \mathbb{E}[\|\theta_t - \theta_{t-\tau}\|]$$
$$\leq d_{TV}(\mathbb{P}(s_{t-1}, a_{t-1} \in \cdot | s_{t-\tau+1}, \theta_{t-\tau}), \mathbb{P}(\tilde{s}_{t-1}, \tilde{a}_{t-1} \in \cdot | s_{t-\tau+1}, \theta_{t-\tau})) + \frac{1}{2} L_\pi \mathbb{E}[\|\theta_t - \theta_{t-\tau}\|]$$

Here $L_\pi$ is the constant that satisfies $\|\pi_\theta - \pi_{\theta'}\| \leq L_\pi \|\theta - \theta'\|$. Repeat the above argument from $t$ to $t - \tau + 1$, we have

$$d_{TV}(\mu_t, \tilde{\mu}_t) \leq \frac{1}{2} L_\pi \sum_{k=t-\tau}^{t} \mathbb{E}[\|\theta_k - \theta_{t-\tau}\|]$$
$$\leq (L_\pi L_g) \tau(\tau+1) \beta_{t-\tau}.$$

Then we turn to bound $d_{TV}(\tilde{\mu}_t, \mu_{\theta_t})$ conditioned on $s_{t-\tau+1}, \theta_{t-\tau}$. By the uniform ergodicity Assumption 1, it shows that

$$d_{TV}(\tilde{\mu}_t, \mu_{\theta_t}) \leq m \rho^{\tau-1}.$$

Thus we can conclude the distance between the Markovian sample and i.i.d, sample is bounded by

$$d_{TV}(\mu_t, \mu_{\theta_t}) \leq (L_\pi L_g) \tau(\tau+1) \beta_{t-\tau} + m \rho^{\tau-1}.$$

We can choose $\tau_T := \min i \geq 0 \mid m \rho^{i-1} \leq \frac{1}{\sqrt{T}}$. Therefore, we choose $\tau_T = \frac{\log m \rho^{-1}}{\log \rho^{-1}} + \frac{\log T}{2 \log \rho^{-1}} = O(\log T)$.

Thus the total variant can be bounded by $O(\log^2(T)/\sqrt{T})$ if we select the step size $O(1/\sqrt{T})$. This associates with the upper bound of the norm of $\|\Delta_t\|, \|\nabla_t\|, \|F(\theta_t)\|$ is the unique term that we need to control, which adds additional $O(\log^2 T/\sqrt{T})$ term in the error term.

We can then give the new theorem for attaining stationary point and global optimum:

**Theorem 3.** *Suppose Assumptions 1 - 5 hold. By choosing step sizes $\alpha_t = c_1/\sqrt{t}$, $\beta_t = c_2/\sqrt{t}$, $\zeta_t = c_3/\sqrt{t}$, where $c_1, c_2, c_3$ are appropriate constants chosen depending on the problem parameters, the sequence of iterates produced by single-loop single-timescale NAC satisfies*

$$\frac{2}{T} \sum_{t=T/2}^{T-1} \mathbb{E}[\|e_t\|^2] \leq O\left(\delta^2 + \frac{\log^2 T}{\sqrt{T}}\right),$$

$$\frac{2}{T} \sum_{t=T/2}^{T-1} \mathbb{E}\left[\|\nabla_t\|^2\right] \leq O\left(\delta^2 + \frac{\log^2 T}{\sqrt{T}}\right),$$

$$\frac{2}{T} \sum_{t=T/2}^{T-1} \mathbb{E}\left[\|\Delta_t\|^2\right] \leq O\left(\delta^2 + \frac{\log^2 T}{\sqrt{T}}\right),$$

*where all parameters except $\delta, T$ are treated as constants in the $O(\cdot)$ notation.*

**Theorem 4.** *Suppose Assumptions 1 - 5 hold. By choosing step sizes $\alpha_t = c_1/\sqrt{t}$, $\beta_t = c_2/\sqrt{t}$, $\zeta_t = c_3/\sqrt{t}$, where $c_1, c_2, c_3$ are appropriate constants chosen depending on the problem parameters, the value functions $V(\theta_t)$ produced by single-loop single-timescale NAC satisfies*

$$V^* - \frac{2}{T} \sum_{t=T/2}^{T-1} \mathbb{E}\left[V(\theta_t)\right] \leq O\left(\delta^2 + \delta' + \log T/T^{1/4}\right),$$

*where all parameters except $\delta', \delta, T$ are treated as constants in the $O(\cdot)$ notation.*

The analysis for these two theorems is both similar with the i.i.d. sampling algorithm except that additional term $\|\cdot\| d_{TV}(\mu_t, \mu_{\theta_t})$ appears, where $\|\cdot\|$ varies from $\|\Delta_t\|, \|\nabla_t\|, \|F(\theta_t)\|$ when controlling these three terms, and these three norms are all bounded by some constants. Then we can get the above two theorems, which state the $\tilde{O}(\epsilon^{-2})$ sample complexity to attain the stationary point and $\tilde{O}(\epsilon^{-4})$ sample complexity to attain the global optimum.

