# OpenReview forum: "Finite Sample Analysis for Single-Loop Single-Timescale Natural Actor-Critic Algorithm"
_ICLR.cc/2024/Conference — ICLR 2024 Conference Withdrawn Submission_

### Official Review · Reviewer_FSCA · 2023-10-16

**Soundness:** 3 good
**Presentation:** 3 good
**Contribution:** 1 poor
**Rating:** 3
**Confidence:** 3

**Summary:**

This paper studies convergence rate for single-loop single-timescale NAC algorithms with linear function approximation.
The authors adapt a moving average style update rule to estimate fisher information and show that with appropriate choices of learning rates, the algorithm takes $O(\epsilon^{-2})$ samples to find an $\epsilon$-approximate stationary points and $O(\epsilon^{-4})$ samples to find an $\epsilon$-global optimal policy.
The authors also provide proof sketch to highlight how to control the error of updates of actor, critic, and fisher information matrix.

**Strengths:**

The paper writing is clear, and easy to follow. The comparison with previous results are also clear. The analysis in Sec. 5.4 is also interesting.

**Weaknesses:**

1. The authors omit the dependence of parameters other than $\epsilon$ in the convergence rate results, which are also very important. I would like the authors to discuss about it, especially, the dependence on feature dimension $d$, $\lambda$ in Assumption 1, the minimal singluar value of $A_{\theta_t}$ (or say $\mu$ in Assumption 5).
Besides, I'm also curious if there is any dependence on something like $\frac{1}{\min_s \mu_{\theta}(s)}$, i.e. the probability to reach some states.

2. Assumption 5 is quite unnatural to me, especially $A_{\theta_t}$ is a non-symmetric matrix. Although the authors claim that it standard, I wonder if the authors could provide some concrete examples?
It seems that this assumption is important to make sure the some places in the proofs can go through, which implies the results are kind of limited given that the assumption can be very hard to verify in practice.

3. The authors motivates the single-loop methods is because it is "more practical" comparing with double-loop methods. I would suggest the authors provide some experiments (even in toy example) to compare with previous double-loop methods.

4. The authors claim that the main challenge is that "the estimation errors of the Fisher information matrix, critic, and the policy gradient are strongly coupled". However, it seems to me that this challenge is kind of artificial. Actually, in Algorithm 4, it is possible to generate three independent $(s,a)$ pairs from $\mu_{\pi_\theta}$ and update $F_t,\theta_t,w_t$ with one of those samples, and there will be no correlation between updates of actor, critic and fisher matrix. Moreover, such process will only increase the sample complexity by a factor of 3, which is constant.

5. Others

* It seems the $\lambda$ denotes regularization coefficient in the update rule of $\theta_t$ while denotes mixing parameter in Assumption 1, please avoid that.

**Questions:**

1. What are the dependence on the other key parameters? (also see Weakness 1)

2. Is there any practical examples so that Assumption 5 is true (even if it is a toy example)? How can we verify it in practice? What results still hold if Assumption 5 is violated? (also see Weakness 2)

---

> ### Author Response · Authors · 2023-11-21
> **Response to Reviewer FSCA**
>
> Thank you for the insightful comment! We will address your concerns as follows:
>  1.  Assumption 5 is a standard assumption which has been commonly assumed in the literature. The assumption holds if the policy \pi_\theta can explore all state-action pairs in the tabular case. For example, if we assume $\pi(a|s) > \epsilon$, then assumption 5 holds with $\mu = \epsilon$.
>  2.  Dependence on key parameters: the dependence on \mu is $O(1/\mu^2)$, and the dependence on $\lambda$ is $O(T^{-1/2} \lambda^2 + \lambda^{-1})$
>  3.  We want to note that the hardness of handling actor, critic, and fisher matrix is not from the correlation of (s_t, a_t) pairs, but from the update form of these three objects. They all need to update themselves using the other two estimated values. Then if one estimation error is large will cause the other two terms to explode too. Relationships of these three terms are shown in Lemma 3, 4, 5. These coupled problem exist even if we use three independent (s, a) pairs to update.
> 4. The assumption that the algorithm uses the i.i.d. sample to update at each step has been replaced by a more realistic Makovian sample in the updated version (pages 22 - 24). This helps the algorithm be more practical. A refined analysis is also shown in the updated version.

---

### Official Review · Reviewer_rbim · 2023-10-26

**Soundness:** 2 fair
**Presentation:** 4 excellent
**Contribution:** 3 good
**Rating:** 5
**Confidence:** 5

**Summary:**

The authors propose a natural actor-critic algorithm to solve the RL problem in discrete state and action spaces. An estimation for the Fisher information matrix using only one sample per step is proposed, which enables the algorithm to have single loop. The authors analyze the convergence property under a single time scale.

**Strengths:**

The algorithm is very close to an online method, except that the samples are drawn from the stationary distribution, which is unknown. Therefore, it could be modified to a practical online algorithm easily.
The analysis for multi-level optimization is usually harder with single time scale, where error terms have interactions that cannot be ignored.
Natural policy gradient usually gives better performance than a vanilla policy gradient, so its theoretical analysis is more meaningful.

**Weaknesses:**

I find some mistakes and gaps in the proof, which I cannot fix myself. The details are in Problems. I would suggest the paper be published if the authors could fix them.
This work is only for linear parametrization of the Q function.

**Questions:**

Major problems:
1.	The proof for Lemma 2 is missing. As a consequence, the proof for Lemma 3 is very hard to understand. In the long equation on Page 14, I can roughly understand that the authors are using Cauchy-Schwartz inequality to bound the terms. But many steps are skipped and there are undefined constants, which make the proof hard to track. Maybe the details are in the proof for Lemma 2?
2.	The proof for Lemma 5. On page 17, I cannot derive the first inequality, where the square of the sum of three terms are bounded by the sum of three separate squares with some coefficients. I think for the last term |F(\theta_{t+1}) – F(\theta_t)|^2, there should be some coefficient like 1/\zeta_t, which will make the proof harder.

Minor problems or typo:
Page 1. Please give full name for TRPO and PPO when they first appear.
Page 1 bottom. “Existing works analyze the single-timescale…” It is better to add some reference for this sentence.
Page 2 Related work first paragraph. When introducing actor-critic method for LQR problem, I think the papers like “Single Timescale Actor-Critic Method to Solve the Linear Quadratic Regulator with Convergence Guarantees” (JMLR 2023) and “Global convergence of two-timescale actor-critic for solving linear quadratic regulator” (AAAI 2023) can be added.
Page 3,6,9. The authors mention that the natural gradient descent is invariant to the parametrization of policies. I think more explanation is needed for this statement. I think the paper “Natural Actor-Critic” (Neurocomputing 2008) gives a good explanation, but I am not sure if this is what the authors mean.
Page 3 about \mu_theta. The definition of \mu_theta is not a distribution, but the visitation frequency. It is better to add a factor (1-\gamma). Also, I don’t think \mu_theta depend on a.
Page 3 and later. For the update of \theta_t, should use + instead of -?
Page 4 It is better to clarify that P_\Omega is the projection operator.
Page 5. According to Assumption 5, does d have to be |A|x|S|?
Page 5. What does “it is known that A is positive definite” mean?
Page 7-8. Some of the terms in the lemmas lack expectation, such as last term in Lemma 2, first term (on the right) in Lemma 3 and 5.
Page 7 Lemma 3. Is the range for “\alpha_t <= \mu/(2L^2)” from T/2 to T?
Page 12. In the definition of \hat{g}, should use Q instead of Q_\theta?
Page 14 first equation. On the second line, there should not be “-\omega_{\theta_{t+1}}”.

---

> ### Author Response · Authors · 2023-11-21
> **Response to Reviewer rbim**
>
> Thank you for the insightful comment! We will address your concerns as follows:
> 1.  We have included the proof of Lemma 2 in the updated version (page 12). This lemma is used to induce the recursion relation of the critic update and Lemma 3 relies highly on Lemma 2. We are sorry for the inconvenience.
>  2.  Sorry for the confusion of the proof of Lemma 5. The proof is carefully refined and updated in the submission (page 18). Additional term $\langle F_t - F(\theta_t), F(\theta_{t+1}-F(\theta_t)) \rangle$ is added in the right side of inequality but it does not affect the $O(\epsilon^{-2})$ order.
>  3.  Thanks for pointing out plenty of minor problems and typos, we have fixed it in the updated version. Additionally, the assumption that the algorithm uses the i.i.d. sample to update at each step has been replaced by a more realistic Makovian sample in the updated version (pages 22 - 24). This helps the algorithm be more practical.

---

> > ### Comment · Reviewer_rbim · 2023-11-22
> > **Thank you for the revision, the paper looks clearer.**
> >
> > Thank you for the revision, the paper looks clearer.

---

### Official Review · Reviewer_qKxN · 2023-10-30

**Soundness:** 2 fair
**Presentation:** 2 fair
**Contribution:** 2 fair
**Rating:** 3
**Confidence:** 3

**Summary:**

This work analyzes the single-loop NAC algorithm with linear function approximation and reaches the $\mathcal{O}(\epsilon^{-4})$ convergence rate.

**Strengths:**

This paper develops a single-loop NAC algorithm, which is new. And moreover shows an $\epsilon^{-4}$-order convergence rate.

**Weaknesses:**

1. The algorithm is designed to be single-loop, yet in Step 3 of Alg1, a sample from the stationary distribution is required. If the authors need to obtain this sample by using some simulator or generator, then I think the novelty here is limited. One of the major problem in single-trajectory algorithm is that the samples are Markovian and cannot be generated as desired, which is bypassed in this paper by using  this generetor.

2. The proof part is hard to read, e.g.,  the proof of Lemma 5, which is the major contribution part. I am not clear about how the first inequality in proof of Lemma 5 can be obtained directly.

3. For AC algorithm analysis, single-loop and double loop AC both can get $\epsilon^{-2}$ convergence rate. In [1], double loop NAC can get $\epsilon^{-3}$ rate. However, this paper only gets $\epsilon^{-4}$ rate, which does not match the SOTA.

**Questions:**

Can you address the ones in the Weaknesses part?

---

> ### Author Response · Authors · 2023-11-21
> **Response to Reviewer qKxN**
>
> Thank you for the insightful comment! We will address your concerns as follows:
>  1.  Thanks for pointing out the limitations of our work. Indeed this assumption of sampling from stationary distribution can be replaced by a more realistic Markovian sampling and only an additional $O(\log^2 T)$ term is added. The proof detailed is shown in the updated version (page 22 - 24, Appendix B).
>  2.  Sorry for the confusion of the proof of Lemma 5. The proof is carefully refined and updated in the submission (page 18). Additional term $\langle F_t - F(\theta_t), F(\theta_{t+1}-F(\theta_t)) \rangle$ is added in the right side of inequality but it does not affect the $O(\epsilon^{-2})$ order.
>  3.  Though there is an O(\epsilon^{-1}) gap between the SOTA global optimum, it is worth noting that we are the first to derive the global optimal guarantee for the single-timescale algorithm. Moreover, the $O(\epsilon^{-2})$ sample complexity to obtain the stationary point matches the SOTA. The key challenge in attaining the $O(\epsilon^{-3})$ global optimum exsits in the term $\sum_{t=0}^{T-1} \sqrt{ \mathbb{E}[\|\omega_t - \omega_{\theta_t} \| ]^2 }$. In the double loop algorithm, one can choose the $O(1/\epsilon^2)$ length for inner loop to obtain $\mathbb{E}[\|\omega_t - \omega_{\theta_t} \| ]^2 < \epsilon^2$ and $O(1/\epsilon)$ length for outer loop. This leads to the final $O(\epsilon^{-3})$ sample complexity. However, it is important to note that they can not guarantee $O(\epsilon^{-2})$ sample complexity to obtain the stationary point and $O(\epsilon^{-3})$ sample complexity to obtain the global optimum at the same time. We use the same time-step to hold the two bounds simultaneously.

---

### Official Review · Reviewer_VKBv · 2023-11-02

**Soundness:** 3 good
**Presentation:** 2 fair
**Contribution:** 2 fair
**Rating:** 5
**Confidence:** 4

**Summary:**

This paper analyzes a single-timescale, single-sample version of the natural actor-critic (NAC) algorithm. The analysis builds on two lines of recent work: (1) finite-time analyses of single-timescale, single-sample actor-critic, and (2) finite-time analyses of natural actor-critic. The paper provides two main results for single-timescale, single-sample NAC: (Theorem 1) an $O(\epsilon^{-2})$ sample complexity for (approximately) obtaining a stationary point of the discounted reward objective; (Theorem 2) an $O(\epsilon^{-4})$ sample complexity for achieving global optimality (up to errors resulting from the actor and critic approximations). The work provides the first analysis of single-timescale, single-sample NAC. Theorem 1 matches the sample complexity of the two-timescale mini-batch NAC analyzed in [Xu et al., 2020] and Theorem 2 falls short of the $O(\epsilon^{-3})$ sample complexity of two-timescale mini-batch NAC in [Xu et al., 2020]. Like [Olshevsky & Gharesifard, 2022] (but unlike [Xu et al., 2020]), the analysis relies on the assumption that samples are drawn directly from the stationary distribution of the current policy at each timestep.

**Strengths:**

The paper provides an analysis of single-timescale, single-sample NAC, addressing a gap that currently exists in the literature. This is likely of interest to the theoretical RL community. The primary novelty in the analysis (cf., Theorem 1 and Sec. 5) is the incorporation of the Fisher information matrix estimation error into existing expressions for the critic and actor errors, and subsequent analysis as an interconnected system of inequalities (reminiscent of the techniques considered in [Chen & Zhao, 2023], [Wu et al., 2020], [Olshevsky & Gharesifard, 2022]). Augmenting existing interconnected system analyses to accommodate the Fisher information appears to require a non-trivial amount of additional algebraic manipulation.

**Weaknesses:**

Though this work addresses an existing gap in the literature, there are significant weaknesses:
1. It is assumed in the analysis that the algorithm samples state-action pairs from the stationary distribution induced by the current policy, $\pi_{\theta_t}$, at each timestep (see Remark 2, page 4). This is in keeping with [Olshevsky & Gharesifard, 2022], but it is an unrealistic assumption. Furthermore, many recent works -- [Chen & Zhao, 2023], [Wu et al., 2020], [Xu et al., 2020] from the references, as well as [Suttle et al., *Beyond exponentially fast mixing in average-reward reinforcement learning via multi-level Monte Carlo actor-critic*, ICML 2023] -- provide analyses that do not require this assumption. The fact that the analyses provided in this paper only match (Theorem 1) or underperform (Theorem 2) existing results that do not require the stationary sampling assumption undermines the significance of the results.
2. It is difficult to understand from the presentation provided in the paper how the analysis provided is related to existing analyses. The proof of Theorem 1 appears to be based on [Olshevsky & Gharesifard, 2022], potentially with influence from [Chen & Zhao, 2023] and others, while the proof of Theorem 2 appears to borrow heavily from the analogous result in [Xu et al., 2020]. However, these relationships are not clearly discussed, making it very difficult to accurately evaluate the contribution of the paper.
3. The fact that the analyses provided in this paper only match (Theorem 1) or underperform (Theorem 2) existing results indicates that there is some "slack" in the kind of analysis used, beyond the stationary sampling assumption. A clear explanation of the reasons for this slackness would be helpful, as it would clarify that limitations of the analysis and provide scope for future work.
4. If the relationship of the analysis to previous work described in 2 above is accurate, then the main novelty in the analysis appears to be handling the introduction of Fisher information matrix estimation error in Theorem 1 (also see first sentence of second paragraph in the conclusion). Though potentially laborious, this would boil down to reworking the algebra of existing analyses, which would be a minor contribution. A more thorough discussion of the key innovations in the analysis is needed to help the reader better understand the significance of this work.

**Questions:**

Additional questions:
* it's stated in the introduction that two-timescale methods artificially slow the actor and thus convergence; in the analysis presented, the actor stepsize must be chosen small enough for the Fisher information matrix estimation error to remain small; how do the actor stepsizes you use compare to those used in two-timescale analyses?
* how does the Fisher information matrix regularization term, $\lambda I$, in the actor update (page 4) affect convergence in Theorem 1 and 2?
* how does the "parameter invariant property of the NPG update" mentioned at the beginning of the paragraph before Theorem 2 (page 6) ensure global optimality in Theorem 2?
* what previous works are the analyses provided in Theorems 1 and 2 based on? what are the key points of departure?

---

> ### Author Response · Authors · 2023-11-21
> **Response to Reviewer VKBv**
>
> Thank you for the insightful comment! We will address your concerns as follows:
> 1.  Though there is an O(\epsilon^{-1}) gap between the SOTA global optimum, it is worth noting that we are the first to derive the global optimal guarantee for the single-timescale algorithm. Moreover, the $O(\epsilon^{-2})$ sample complexity to obtain the stationary point matches the SOTA. The key challenge in attaining the $O(\epsilon^{-3})$ global optimum exsits in the term $\sum_{t=0}^{T-1} \sqrt{ \mathbb{E}[\|\omega_t - \omega_{\theta_t} \| ]^2 }$. In the double loop algorithm, one can choose the $O(1/\epsilon^2)$ length for inner loop to obtain $\mathbb{E}[\|\omega_t - \omega_{\theta_t} \| ]^2 < \epsilon^2$ and $O(1/\epsilon)$ length for outer loop. This leads to the final $O(\epsilon^{-3})$ sample complexity. However, it is important to note that they can not guarantee $O(\epsilon^{-2})$ sample complexity to obtain the stationary point and $O(\epsilon^{-3})$ sample complexity to obtain the global optimum at the same time. While we use the same time-step to hold the two bounds simultaneously.
> 2.  We thank the reviewer for pointing out that the algorithm samples state-action pairs from the stationary distribution induced by the current policy is a very strong assumption. Indeed we have removed this assumption and given the refined analysis in the updated PDF (Appendix B). Additional $O(\log ^2 T)$ order is added to attain the stationary point and  $O(\log T)$ order is added to attain the global optimum.
> 3.  The reason for obtaining the global optimality for natural policy gradient is that we can show from the performance difference lemma that the difference between the optimal value function V^\ast and V^t can be bounded by the  KL divergence between the optimal policy and current policy, which is directly optimized by NPG rather than PG. Thus we can then show the monotonic performance improvement of NPG and the global optimality. The analysis builds on [1] and we extend it to the NAC analysis.
> 4.  Relationship with prior work: We want to clarify that the recursion relation of the critic update is inspired by [Olshevsky & Gharesifard, 2022], and the analysis to obtain the global optimum is based on the two-timescale analysis for NAC in [Xu et al., 2020]. However, it is important to note that the analysis for actor update and the estimated fisher information matrix is novel in this paper. The analysis to attain the stationary point by handling the three deeply coupled terms is also new and non-trivial compared to [Olshevsky & Gharesifard, 2022], which only handles two coupled terms.
> 5.  The design \lambda I in the actor update is to prevent the estimated Fisher information matrix from being non-singular. In the Theorem 1 is of order $O(1 / \lambda^2)$. In Theorem 2, the dependence on \lambda is $O(T^{-1/4}+T^{-1/2} \lambda^{-2} + \lambda)$.
> 6.  The actor stepsize we choose is $O(1 / \sqrt{t})$, which is in the same order as the critic update and Fisher information matrix update. In the two timescale update, the actor stepsize is chosen as $O(1 / t^{3/5})$ and the critic stepsize is $O(1 / t^{2/5})$. Note that the choice of  $O(1 / \sqrt{t})$ is standard in the literature of online learning such as the online mirror descent algorithm and follows the regularized leader algorithm.
>
> [1] Xuyang Chen and Lin Zhao. Finite-time analysis of single-timescale actor-critic. arXiv preprint arXiv:2210.09921, 2022.

---

> > ### Comment · Reviewer_VKBv · 2023-11-23
> >
> > Thanks to the authors for their responses. Parts 1, 3, 5, and 6 are clarifying. Regarding 2, I appreciate the attempt in the new Section B to adapt [Chen & Zhao, 2022] to eliminate the stationary sampling assumption. However, the total variation bound must be propagated throughout the analysis to ensure that the occupancy measure "drift" arising from the changing policy parameters and the consequent biases in the policy gradient, Fisher information, and critic estimates remain manageable. Regarding 4, though I sincerely appreciate the additional discussion of the relationship to existing work, I am still unclear exactly what key innovations over previous works are required in the analysis. I remain concerned that, as pointed out in part 4 of the Weaknesses section in my review, the primary effort involved a simple reworking of the algebra in the actor-critic analysis to handle the Fisher information matrix estimation. In light of these remaining issues, I will keep my score.